



# Box canyon erosion along the Canterbury coast (New Zealand): A rapid and episodic process controlled by rainfall intensity and substrate variability

Aaron Micallef[1,2], Remus Marchis[3], Nader Saadatkhah[1], Roger Clavera-Gispert[1], Potpreecha Pondthai[4], Mark E. Everett[4], Anca Avram[5,6], Alida Timar-Gabor[5,6], Denis Cohen[2], Rachel Preca Trapani[2], Bradley A. Weymer[1]

[1]Helmholtz Centre for Ocean Research, GEOMAR, Kiel, Germany
[2]Marine Geology & Seafloor Surveying, Department of Geosciences, University of Malta, Malta
[3]Department of Geological Sciences, University of Canterbury, Christchurch, New Zealand
[4]Department of Geology and Geophysics, Texas A&M University, Texas, USA
[5]Faculty of Environmental Science and Engineering, Babes-Bolyai University, Cluj-Napoca, Romania
[6]Interdisciplinary Research Institute on Bio-Nano-Sciences, Babeş-Bolyai University, Cluj-Napoca, Romania

*Correspondence to*: Aaron Micallef (amicallef@geomar.de)

**Abstract:** Box canyon formation has been associated to groundwater seepage in unconsolidated sand to gravel sized sediments. Our understanding of box canyon evolution mostly relies on experiments and numerical simulations, and these rarely take into consideration contrasts in lithology and permeability. In addition, process-
based observations and detailed instrumental analyses are rare. As a result, we have a poor understanding of the temporal scale of box canyon formation and the influence of geological heterogeneity on their formation. We address these issues along the Canterbury coast of the South Island (New Zealand) by integrating field observations, optically stimulated luminescence dating, multi-temporal Unmanned Aerial Vehicle and satellite data, time-domain electromagnetic data, and slope stability and landscape evolution modelling. We show that box
canyon formation is a key process shaping the sandy gravel cliffs of the Canterbury coastline. It is an episodic process associated to groundwater flow that occurs once every 227 days on average, when rainfall intensities exceed 40 mm per day. The majority of the box canyons in a study area SE of Ashburton has undergone erosion, predominantly by elongation, during the last 11 years, with the most recent episode occurring 3 years ago. The two largest box canyons have not been eroded in the last 2 ka, however. Canyons can form at rates of up to 30 m
per day via two processes: the formation of alcoves and tunnels by groundwater seepage, followed by retrogressive slope failure due to undermining and a decrease in shear strength driven by excess pore pressure development. The location of box canyons is determined by the occurrence of hydraulically-conductive zones, such as relict braided river channels and possibly tunnels, and of sand lenses exposed across sandy gravel cliff. We also show that box canyon formation is best represented by a linear diffusive model and geometrical scaling.



## 1    Introduction

Whereas erosion by overland flow has been extensively studied and associated geomorphic rate laws have been developed (Sklar and Dietrich, 2001; Whipple et al., 2000), erosion by groundwater flow has received considerably less attention. Groundwater can mechanically erode or deform sediment and bedrock by removal of mass from a seepage face through flowage, frictional slipping and particle dislodgement (Dunne, 1990; Iverson and Major, 1986). Groundwater seepage can also lead to slope instability by undercutting, reduction of soil shear

strength and shear surface development (Carey and Petley, 2014; Chu-Agor et al., 2008). Groundwater has been implicated as an important geomorphic agent in valley network development, both on Earth and on Mars (Abotalib et al., 2016; Dunne, 1990; Harrison and Grimm, 2005; Higgins, 1984; Kochel and Piper, 1986; Malin and Carr, 1999; Salese et al., 2019). For over one hundred years, a characteristic morphology - comprising theatre-shaped heads, steep and high valley walls, constant valley width, flat floors, low drainage densities, and short tributaries

with large junction angles - has been cited as diagnostic of groundwater activity (Abrams et al., 2009; Dunne, 1990; Higgins, 1984; Russel, 1902; Schumm and Phillips, 1986). We refer to valleys with these characteristics as box canyons.

The classic model explaining the development of box canyons entails a canyon headwall that lowers the local

hydraulic head and focuses groundwater flow to a seepage face. This leads to upstream erosion by undercutting, the rate of which is limited by the capacity of seepage water to transport sediment from the seepage face (Abrams et al., 2009; Dunne, 1990; Howard and McLane, 1988). Groundwater seepage has been shown to unambiguously lead to box canyon formation in unconsolidated sand to gravel sized sediments (Dunne, 1990; Lapotre and Lamb, 2018) (Table 1). In sediments finer than sands, erosion is typically limited by detachment of the grains at the

seepage face. In silts and clays, the permeability is so low that the groundwater discharge is often less than that required to overcome the cohesive forces of the grains (Dunne, 1990). The role of groundwater seepage and box canyon formation in bedrock, on the other hand, remains controversial (Lamb et al., 2006; Pelletier and Baker, 2011).

**Table 1: Examples of studies documenting box canyons attributed to groundwater seepage erosion in unconsolidated sediments (sand- to gravel-sized).**

| Location | Substrate type | Reference |
|---|---|---|
| Alaska, USA | Gravel braided river deposits | (Sunderlin et al., 2014) |
| Canterbury Plains, New Zealand | Gravel braided river deposits | (Schumm and Phillips, 1986) |
| Florida Panhandle, USA | Non-marine quartz sands that contain discontinuous layers of clay or gravel | (Schumm et al., 1995) |
| Kalahari, southern Africa | Conglomerates, marls, duricrusts and unconsolidated sands | (Nash et al., 1994) |
| Martha's Vineyard and Nantucket Island, USA | Outwash gravelly sand | (Uchupi and Oldale, 1994) |
| Obara area, Japan | Granodiorite regolith | (Onda, 1994) |
| South Taranaki, New Zealand | Terrestrial dune sand and tephra overlying marine sands and gravels | (Pillans, 1985) |
| Vocorocas, Brazil | Alluvial sands | (Coelho Netto et al., 1988) |



Our understanding of box canyon evolution is predominantly derived from theoretical, experimental and numerical models, which have reproduced the characteristic box canyon morphologies in unconsolidated

sediments (Chu-Agor et al., 2008; Higgins, 1982; Howard, 1995; Lobkovsky et al., 2004; Pelletier and Baker, 2011; Petroff et al., 2011; Wilson et al., 2007). Such studies suggest that the velocity at which box canyon heads advance is a function of groundwater flux and the capacity of seepage water to transport sediment from the seepage face (Abrams et al., 2009; Fox et al., 2006; Howard, 1988; Howard and McLane, 1988), and that canyon head erosion occurs by episodic headwall slumping (Howard, 1990; Kochel et al., 1985). Streams incised by

groundwater seepage have been shown to branch at a characteristic angle of 72° at stream tips, which increases to 120° near stream junctions (Devauchelle et al., 2012; Yi et al., 2017), whereas growing indentations competing for draining groundwater result in periodically-spaced valleys (Dunne, 1990; Schorghofer et al., 2004). Canyon network geometry appears to be determined by external groundwater flow field rather than flow within the canyons themselves (Devauchelle et al., 2012).


A number of fundamental questions related to the evolution of box canyons in unconsolidated sediments remain unanswered. Firstly, the temporal scale at which box canyons form is poorly quantified due to a paucity of process-based observations and detailed instrumental analysis. Field observations of groundwater processes are rare (e.g. Onda, 1994), primarily due to the difficulty with accessing the headwalls of active canyons, the potential long

timescales involved, and the complexity of the erosive process (Chu-Agor et al., 2008; Dunne, 1990). Quantitative assessments of box canyon evolution have relied on experimental and numerical analyses, but these tend to be based on simplistic assumptions about flow processes and hydraulic characteristics. Experimental approaches are based on a range of different methods, which limits comparison of their outcomes (Nash, 1996). Published erosion rates vary between 2-5 cm per century (Abrams et al., 2009; Schumm et al., 1995) and 450-1600 $m^3$ per year

(Coelho Netto et al., 1988). Secondly, the influence of geologic heterogeneities on box canyon evolution is also poorly understood. Lithological strength and permeability contrasts are rarely simulated by experimental and numerical analyses. Thirdly, there only a few places where the mechanisms by which seepage erosion occurs have been clearly defined (e.g. Abrams et al., 2009). Basic observations and measurements of box canyon erosion rates and substrate geologic heterogeneities are needed to test and quantify models for canyon formation and improve

our ability to reconstruct and predict landscape evolution by groundwater-related processes.

In this study we revisited the Canterbury Plains study site (Schumm and Phillips, 1986) and carried out field observations, geochronological analyses, repeated remote sensing surveys, near-surface geophysical surveying and numerical modelling of coastal box canyons to: (i) identify the processes by which groundwater erodes box

canyons along the coast, (ii) quantify the timing of box canyon erosion and its key controls, and (iii) assess the influence of geological/permeability heterogeneity on the box canyon formation process.

## 2    Regional setting

The flat to gently inclined Canterbury Plains, located in the eastern South Island of New Zealand, extend from sea level up to 400 m above sea level, and cover an area 185 km long and 75 km wide (Fig. 1a). A series of high energy braided rivers emerge from the >3500 m high Southern Alps and flow south-eastwards to the shoreline



(Kirk, 1991). The plains were formed by coalescence of several alluvial fans sourced from the these rivers (Browne and Naish, 2003; Leckie, 2003). The Quaternary sedimentary sequence comprises a >600 m thick

succession of cyclically stacked fluvio-deltaic gravel, sand and mud with associated aeolian deposits and palaeosols (Bal, 1996; Browne and Naish, 2003). The gravels consist of greywacke and represent a variety of channel fill beds and bar forms, whereas the isolated bodies of sand are relict bars and abandoned channels. The interglacial sediments are better sorted and have higher permeability than the glacial outwash, resulting in a wide range of hydraulic conductivities (Scott, 1980). New Zealand's largest groundwater resource is hosted in the

gravels down to at least 150 m depth (Davey, 2006). The upper Quaternary sediments are exposed along a 70 km long coastline south-west of the Banks Peninsula (Moreton et al., 2002). This coastline is retrograding at approximately 0.5–1 m per year and consists of cliffs fringed by mixed gravel and sand beaches (Gibb, 1978). The study area is a 2.5 km long stretch of cultivated coastline located 16 km to the south-east of Ashburton (Fig. 1b). The coastline within the study area consists of a 15-20 m thick exposure of poorly-sorted and uncemented

matrix-supported outwash gravel, which is capped by up to 1 m of post-glacial loess and modern soil (Berger et al., 1996). The cliff face is punctuated by ~0.5 m thick lenses of sand or clean gravel.


Earth **Surface**
Dynamics
Discussions

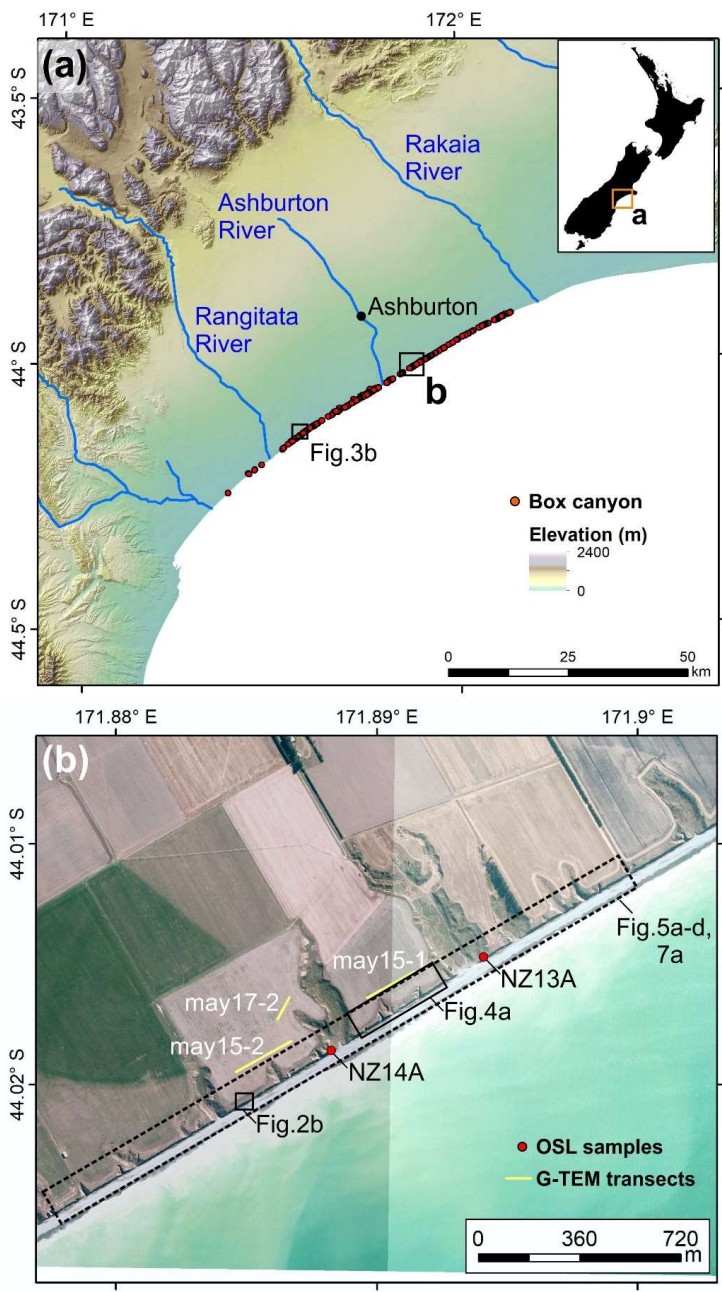

**Figure 1: (a) Digital elevation model of the Canterbury Plains (source: Environment Canterbury), located along the eastern coast of the South Island of New Zealand, showing the location of mapped box canyons. Location of figure is shown in inset. (b) Mosaic of aerial photographs of the study area (source: Environment Canterbury). Location in a. Location of Optically Stimulated Luminescence (OSL) samples, G-TEM transects, and other figures is shown.**



## 3 Materials and Methods

### 3.1 Data

#### 3.1.1 Field visits

Site visits were carried out in May 2017 and 2019. During these visits, geomorphic features of interest were noted and photographed, and samples collected. The latter included outcropping sediment layers across the cliff face, sediments with coating, and loess sediments draping the two largest box canyons for geochronological analysis (NZ13A, NZ14A; Fig. 1b). The latter were collected by hammering stainless steel tubes into the sediment and ensuring that the material was not exposed to light.

#### 3.1.2 Unmanned aerial vehicle (UAV) surveys

UAV surveys were carried out using DJI Phantom 4 Pro and DJI Mavic Pro drones. The drones were flown at an altitude of 40-55 m, speed of 5 m s$^{-1}$, and side lap of 65-70%. Eight ground control points were selected and their location determined by differential GPS. Orthophotos and digital elevation models with a horizontal resolution of 10 cm/pixel were generated from the UAV data using Drone Deploy. The surveys were carried out on the following dates: 11[th] May, 19[th] June, 30[th] June, 11[th] July, 15[th] July, 23[rd] July, 29[th] July, 4[th] August, 26[th] August, 11[th] September, 23[rd] September, 6[th] October, 13[th] October and 30[th] October 2017.

#### 3.1.3 Near-surface geophysics

Time-domain electromagnetic (TEM) measurements were carried out in May 2019 using the Geonics (Canada) G-TEM system (Fig. 1b). The operating principles of the inductive TEM technique are described in Nabighian and Macnae (1991) and Fitterman (2015). The survey parameters included 4 turns, a $10 \times 10$ m$^2$ square TX loop, and a TX current output of 1 A. The G-TEM was operated in a fixed offset-sounding configuration, which is termed "Slingram" mode, in which the RX coil was placed 30 m from the centre of the TX loop and the TX-RX pair moved together along a linear transect at 5 m station spacing, maintaining the 30 m offset. All soundings were collected in the 20-gate mode with an acquisition interval of $6 \times 10^{-6}$ s to $8 \times 10^{-4}$ s (after ramp-off), corresponding to investigation depths of ~80 m. At each station, a consistent 1-D smooth model of electrical resistivity vs. depth was performed based on the iterative Occam-regularised inversion method (Constable et al., 1987) and using the IXG-TEM software from Interpex Limited.

#### 3.1.4 Other data

Satellite images with a horizontal resolution of 1 m/pixel and dating back to 2004 were obtained from Google Earth. Precipitation records dating back to 1927 were provided by Environment Canterbury. The latter also





provided a time series of water level data since 2015 from a 30 m deep well located 10 km to the north-east of the study area.


### 3.2 Methods

#### 3.2.1 Sample analyses

Sediment samples were analysed for grain size distribution using sieves following the ASTM D0422 protocol. The composition of the coating on selected sediment outcrops within the box canyons was determined using X-ray fluorescence.

#### 3.2.2 Optically Stimulated Luminescence (OSL) dating


Luminescence dating is numerical-age technique that uses optically and thermally sensitive signals measured in the form of light emissions in the constituent minerals that form sediment deposits. Quartz and feldspars are among the most often used minerals. Sediment ages obtained via luminescence dating reflect the last exposure of the analysed mineral grains to daylight, when the resetting (called bleaching) of the previously incorporated 190 luminescence signal occurs.

In order to obtain luminescence ages, two types of measurements were performed. The dose accrued by the crystal from natural radioactivity since its last exposure to daylight (called the palaeodose) was determined as an equivalent dose ($D_e$). This was done by measuring the light emission of the crystal upon optical stimulation, and 195 matching this emission to signals generated by the exposure to a known dose of radiation given in the laboratory. This is expressed as the amount of absorbed energy per mass of mineral (1 J kg$^{-1}$ = 1 Gy (Gray)). Radioactivity measurements were carried out on each sample in order to determine the annual dose ($D_a$), which represents the rate at which the environmental dose was delivered to the sample (Gy ka$^{-1}$). The age was obtained by dividing the two determined parameters. As low luminescence sensitivity and poor dosimetric characteristics were reported 200 for quartz from sediments in New Zealand (see Preusser et al. (2009) and the references cited therein) we have used signals from feldspars by the application of infrared stimulation based on the post IR-IRSL$_{225}$ (Buylaert et al., 2009) and post IR-IRSL$_{290}$ (Thiel et al., 2011) protocols on polymineral fine (4-11µm) grains, as well as coarse (63-90 µm) potassium feldspars.

A detailed description of luminescence dating methodology, including sample preparation, equivalent dose determination, annual dose determination, luminescence properties (including residual doses, dose recovery tests and fading tests), is presented as Supplementary Materials.





### 3.2.3 Morphological change detection


The method used to measure box canyon aerial erosion in between surveys entailed the manual delineation of shapefiles around box canyon boundaries for each survey (using orthophotos, digital elevation models and slope gradient maps in the case of the UAV data, and satellite images in the case of the Google Earth data), the estimation

of their areas, and the comparison of the latter in between surveys. The uncertainty inherent in this approach is related to the digitisation of the canyon boundaries. We made sure that a vertex was added at least every 5 pixels for both the UAV and Google Earth data. This ensures that a minimum erosion of 0.25 m² (in the case of the UAV data) and 25 m² (in the case of the Google Earth data) was detected.

### 3.2.4 Modelling


***Slope stability modelling***

We developed a slope stability model based on the limit equilibrium and segmentation strategy of the Bishop

method, where a soil mass is discretised into vertical slices. The factor of safety $F_f$ is calculated using the following (Fredlund and Krahn, 1977; Fredlund et al., 1981):

$$F_f = \frac{\sum \left( c' \beta \cos\alpha + (N - u\,\beta) \tan\phi' \cos\alpha \right)}{\sum N \sin\alpha - \sum D \cos\omega} \tag{1}$$

where $c'$ (in kPa) is effective cohesion, $\phi'$ (in °) is effective angle of friction, $u$ (in kPa) is pore-water pressure, $D$ (in kN) is concentrated point load, $\beta$ (in m) represents geometric parameters, $\omega$ is geometric parameter, and $\alpha$ (in °) is inclination of slice base. $N$ is the normal force acting on the slide base and can be computed by:

$$N = W \cos\alpha - kW \sin\alpha + [D \cos(\omega + \alpha - 90)] \tag{2}$$


where $W$ (in kN) is slice weight (unit weight $\gamma_s$ (in kN m⁻³) × volume (in m³)) and $k$ is hydraulic conductivity (in m s⁻¹).

We also modelled the water flow and pore pressure distribution within the soil using the Poisson equation, which

is the generalised form of the famous Laplace equation (Whitaker, 1986):

$$k_x \frac{\partial^2 h}{\partial x^2} + k_y \frac{\partial^2 h}{\partial y^2} = q \tag{3}$$

where $q$ is the total discharge (m³ s⁻¹), $k_x$ and $k_y$ are equal to the hydraulic conductivity (m s⁻¹) in the horizontal

and vertical directions, respectively, and $h$ is the hydraulic head (m).



Equation (3) applies to water flow under steady-state and homogeneous conditions, whereas the following equation is applicable to dynamic and inhomogeneous conditions:

$$\frac{\partial}{\partial_x}(k_x \frac{\partial h}{\partial x}) + \frac{\partial}{\partial_y}(k_y \frac{\partial h}{\partial y}) = q + \frac{\partial \theta}{\partial t}$$


(4)

where $\partial \theta / \partial t$ describes how the volumetric water content changes over the time.

The water transfer theory accounts for transient behaviour, which can be defined by the following equation (Domenico and Schwartz, 1997):


$$M_{st} = \frac{dM_{st}}{dt} = m_{in} - m_{out} + M_s$$

(5)

where $m_{in}$ is the cumulative mass of water that enters the porous medium, $m_{out}$ is equal to the mass of water that leaves the porous medium, and $M_s$ is the mass source within the representative elementary volume. The rate of increase in the mass of water stored within the representative elementary volume is:


$$M_{st} = M_w + M_v$$

(6)

where $Mw$ and $Mv$ represent the rate of change of liquid water and water vapour, respectively.


The mechanical and hydraulic soil properties employed in this model are listed in Table 2 and were obtained from Dann et al. (2009) and Aqualinc Research Limited (2007). We modelled two scenarios, based on the available rainfall data (see Sect. 4.4). The first is a 3-day long intense rainfall event ($(I-D)_3$) covering the period 20[th] July – 22[nd] July 2017. The second is a 14-day period with occasional, low intensity rain ($(I-D)_{14}$) between the 21[st] June and 4[th] July 2017. Each scenario is modelled for two sandy gravel slopes, one with a 0.5 m thick sand lens and the other with a 0.5 m thick gravel lens. Lateral water inflow and surface water infiltration were estimated from the hydrological model in Micallef et al. (2020). Slope stability modelling was carried out using the Slide2 software package by Rocscience.



**Table 2: Mechanical and hydraulic soil properties used in slope stability modelling.**

| Soil type | Unit weight (kN m$^{-3}$) | Cohesion (kPa) | Friction angle (°) | Saturated hydraulic conductivity (m/day) | Residual water content (m$^3$ m$^{-3}$) | Saturated water content (m$^3$ m$^{-3}$) |
|---|---|---|---|---|---|---|
| | $\gamma_s$ | $c$ | $\varphi$ | $k$ | $\theta r$ | $\theta s$ |
| Sand | 20.5 | 7 | 34.5 | 3.216 | 0.01 | 0.078 |
| Sandy gravel | 23 | 8 | 37 | 0.64 | 0.01 | 0.128 |
| Gravel | 24 | 4 | 36.5 | 7.376 | 0.016 | 0.142 |



***Landscape evolution modelling***

We also built a landscape evolution model (LEM) using the Python modelling environment Landlab (Barnhardt et al., 2020; Hobley et al., 2017). The model allowed us to simulate two main processes: groundwater flow and associated erosion (Fig. 2).

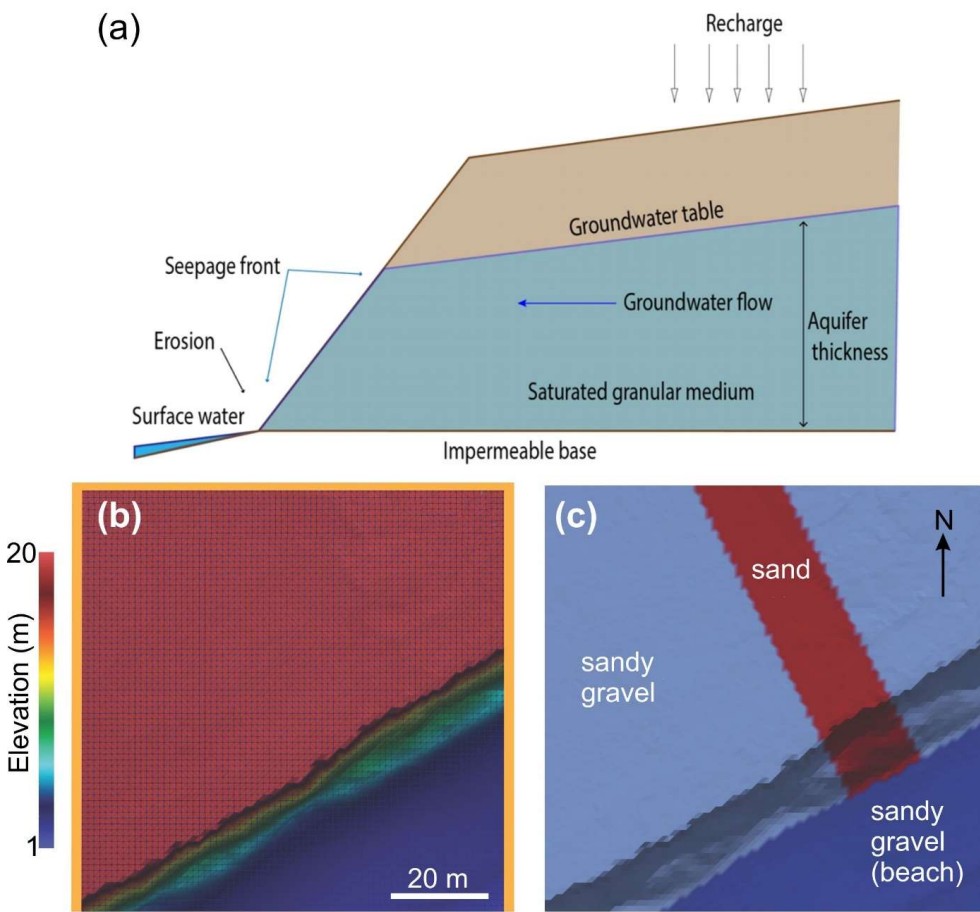

**Figure 2: (a) Conceptual model of the LEM. (b) Initial topography used in the LEM. Orange line denotes closed boundary. (c) Spatial distribution of sediment type in the LEM.**

The model uses the Dupuit-Forchheimer approximation to simulate groundwater flow, which was developed for Landlab by Litwin et al. (2020). This element of the model solves the Boussinesq equation for horizontal flow

direction in an unconfined aquifer over an impermeable aquifer base. For a detailed description of the theory and its implementation in Landlab, the reader is referred to the documentation in https://landlab.readthedocs.io/en/master/reference/components/dupuit_theory.html.   We   computed   the groundwater flow and the groundwater seepage; the latter occurs when the water table intersects the topographic





surface and then becomes surface runoff. The model considers that all surface water is derived from groundwater
seepage.

We used the model to test two erosional modes. The first mode is the stream power law model, which simulates
surface erosion (Barnhart et al., 2019; Braun and Willett, 2013):

$$\frac{\partial \eta}{\partial t} = -KQ^m S^n \tag{7}$$

where $\eta$ is the topographic elevation [L], $t$ is time [T], $K$ is an erosion coefficient [L$^{1-3m}$ T$^{m-1}$], $Q$ is the surface
water discharge [L$^3$T$^{-1}$], $S$ is the slope (dimensionless) and $m$ and $n$ are exponents (dimensionless). We assumed
$m = 0.5$ and $n = 1$, due to a lack of data and a robust methodology to determine these two parameters (Harel et al.,
305    2016).

The second mode is a linear diffusion to simulate erosion by gravitational sediment movement (Barnhart et al.,
2019; Culling, 1963):

$$\frac{\partial \eta}{\partial t} = -D\nabla^2 \eta \tag{8}$$

where $D$ is the diffusion coefficient [L$^3$T$^{-1}$].

For the sake of simplicity and to easily compare the results, $K$ (Eq. (7)) is computed as $D$ (Eq. (8)), which we
consider to be proportional to surface water shear stress and seepage flux:

$$D = Q + M\left(\frac{\tau_b - \tau_t}{\tau_t}\right) \tag{9}$$

where $\tau_b$ is the shear stress [MLT$^{-2}$], $\tau_t$ is the threshold shear stress [MLT$^{-2}$] (assumed to have a value of 0.03 N
m$^{-2}$), and $M$ is an empirical parameter [L$^3$T$^{-1}$] that we assume to have a value of 0.1 m$^3$ s$^{-1}$. The values for $\tau_t$ and
$M$ are empirical and were estimated by trial and error.

Additionally, as a theoretical exercise, we tested the evolution of the surface using the stream power law approach
with surface flow and without groundwater seepage. Equation (7) was changed by including surface water
discharge ($Q$) from an upstream drainage area:

$$\frac{\partial \eta}{\partial t} = -KA^m S^n \tag{10}$$






The initial topography was extracted from a digital elevation model of part of the study area that has not been affected by canyon erosion (Fig. 2b), which allowed us to incorporate a realistic approximation of the cliff morphology. To simulate the groundwater flow, we arbitrarily defined the base of the aquifer at 0.1 m above sea

level, to minimise vertical groundwater movement, and applied the Dupuit-Forchheimer approximation. The initial water table was placed at 5.9 m above sea level (in accordance with well data). Since Landlab is a 2-D modelling environment, it was not possible to include sand lenses. For this reason, we simplified the geology by considering two types of sediment: sandy gravels and with a NW-SE strip of sand. Their distribution and properties are shown in Fig. 2c and Table 2, respectively. In Eq. (10), we assumed that the erosion coefficient ($K$)

of the sand unit is the double that of sandy gravel.

The aquifer recharge was modelled using two different precipitation scenarios. The 'high intensity' scenario is based on a first hour of low recharge ($1 \times 10^{-9}$ m s$^{-1}$) and a 2 hour long high recharge storm ($2.8 \times 10^{-5}$ m s$^{-1}$). The latter value was selected to replicate the rise in the water table reported after the 21$^{st}$ July 2017 storm (see Sect.

4.4.1). The 'low intensity' scenario consists of a first hour of low recharge ($1 \times 10^{-9}$ m s$^{-1}$), followed by a 5 hour long low recharge storm ($1.12 \times 10^{-5}$ m s$^{-1}$). The total modelling time was 24 hours.

### 4        Results

### 4.1        Box canyons along the Canterbury coast - distribution and morphology

We have mapped 315 box canyons (locally also known as "dongas") along 70 km of the Canterbury coastline (mean of 4.5 canyons per km of coastline). The distribution of the box canyons is clustered (nearest neighbour ratio of 0.33 with a z-score of -22.67 and a p-value of 0); the majority of the canyons are located between Rakaia

and Rangitata Rivers (Fig. 1a), particularly in the vicinity of Ashburton River. The heads of many box canyons connect to shallow, relict meandering channels (Fig. 3a). Some of these channels are visible in aerial photographs, in spite of the terrain having been worked by farmers (Fig. 3b).



Earth **Surface**
**Dynamics**
Discussions

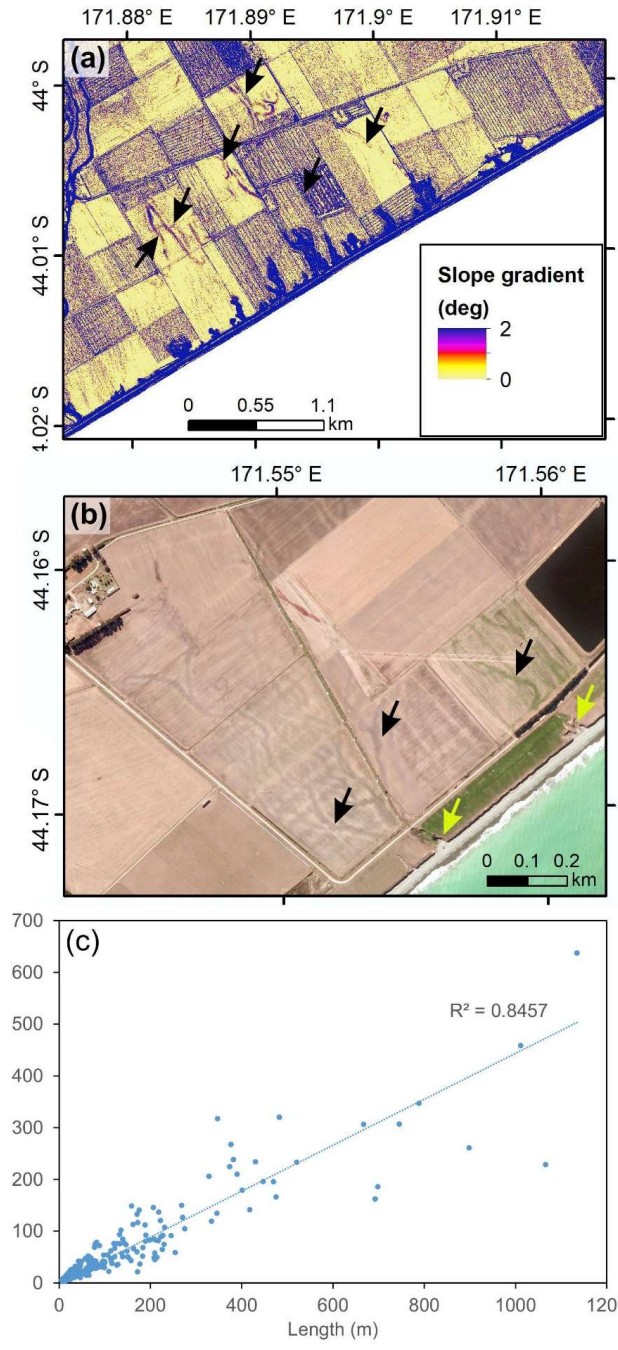


**Figure 3: (a) Slope gradient map of the study area. Black arrows indicate relict, infilled channels. (b) Aerial photograph of Coldstream, Canterbury coast (source: Environment Canterbury). Black arrows indicate relict, infilled channels. Yellow arrows indicate box canyons. Location in Fig. 1a. (c) Plot of length vs. width for box canyons mapped along the Canterbury coastline.**




In plan, the box canyons are predominantly linear to slightly sinuous (sinuosity of 1-1.3) and characterised by a concave canyon head. In profile, the canyons have linear, gently sloping (2-10°) axes, with a concave break of slope separating the axis from a steep (up to 70°) head. In cross-section, the canyons are U-shaped with walls up to 70° in slope gradient. The box canyons are between 5 and 1134 m long (mean of 116 m) and between 3 and

637 m wide (mean of 56 m). Canyons generally exhibit a constant valley width with distance upslope. They have a length to width ratio that varies between 1 and 7.9, with a mean of 2 (standard deviation of 0.89) (Fig. 3c).

### 4.2    Field site observations

In May 2017, our study area hosted 33 box canyons that vary between 15 m and 600 m in length (Figs. 1b; 4a). During the site visits we did not encounter evidence of surface flow. However, the middle to lower sections of the canyon walls and cliffs were consistently wet. These sections were also characterised by failure scars and alcoves, particularly above the sandy lenses. Alcoves were also encountered at the base of canyon heads, where they were wet and up to 1 m deep (Fig. 4e). Some sandy layers outcropping across the cliff face hosted tunnels (Fig. 4c-d).

Above these tunnels, theatre-shaped scars with a shallow and narrow gully at their base were observed (Fig. 4c). At the base of the scars, the box canyon heads and some box canyon mouths, we encountered mass movement debris that was occasionally intact and that predominantly consisted of gravel, sandy gravel and loess (Fig. 4c, h). Box canyons have gravel-covered irregular floors. Whereas the smaller box canyon have a U-shaped cross-section, the larger box canyons have gently sloping V-shaped cross-sections, with loess draping their walls (Fig.

4f). Sandy and clean gravel layers outcropping within the canyons were wet; the former appeared weathered, whereas the latter were coated by Fe and Mn (Fig. 4g). Fences were locally seen suspended across a number of box canyons (Fig. 4b).





**Figure 4: (a) Orthophoto map of part of the study area draped on a 3-D digital elevation model. Location in Fig. 1b. (b-h) Photographs of features of geomorphic interest taken at study area.**



### 4.3 Luminescence ages

Four sets of OSL ages are presented in Table 3. The pIRIR$_{290}$ ages are higher than the ages obtained by applying pIRIR$_{225}$ protocol. The cause of this difference is not yet fully understood, although it can partially be attributed to the results of the dose recovery test and the poor bleachability of the pIRIR$_{290}$ signals compared to pIRIR$_{225}$ signals (Buylaert et al., 2011). Considering that no anomalous behaviour of the investigated signals was observed (see Supplementary Materials), we are unable to explain the overestimation of the K-feldspar ages compared to

the polymineral fine grain ages in the case of NZ13A, especially since the opposite behaviour is observed in the case of sample NZ14A. However, considering a 95% confidence level, ages obtained using different methods broadly overlap, the only exception being the pIRIR$_{225}$ ages obtained on K-feldspars on sample NZ14A, which we regard as an outlier.

**Table 3: Summary of the pIRIR$_{225}$ and pIRIR$_{290}$ ages obtained on polymineral fine grains (4-11 µm) and coarse K-feldspars (63-90 µm). The ages were determined considering 15% water content. Uncertainties are given at 1σ, 68% confidence level. Further details are available in the Supplementary Materials.**

|  | Age (ka)-pIRIR225 | | Age (ka)-pIRIR290 | |
| --- | --- | --- | --- | --- |
| **Sample code** | **Polymineral fine grains** | **K-feldspars (63-90 µm)** | **Polymineral fine grains** | **K-feldspars (63-90 µm)** |
| **NZ13A** | 16.0±1.4 | 20.1±1.5 | 20.9±2.0 | 26.2±2.1 |
| **NZ14A** | 4.6±0.4 | 1.9±0.1 | 6.0±0.7 | 3.1±0.3 |

### 4.4 Morphological changes


#### 4.4.1 Short term morphological changes

By comparing the orthophotos and digital elevation models generated from the UAV data acquired during the various site visits between May and October 2017, we document the formation of 3 new box canyons (up to 30 m

long, Figs. 5e-f) and the enlargement of 30 box canyons (primarily by elongation, and occasionally by widening and branching) (Figs. 5a-d). The new box canyons formed at locations where there was a small landslide scar in the middle of the cliff. There was no change in form in 3 of the box canyons. Figure 6 shows the total area eroded between surveys (which amounts to 3273 m$^2$), the daily precipitation and the associated changes in water table height. Only three surveys recorded box canyon erosion. Two of these surveys happened soon after rainfall events

of >40 mm in one day (Fig. 6). The most important of these covers the period between the 15$^{th}$ and 23$^{rd}$ July 2017, when 95% of the material was removed and the 3 new box canyons were formed (Figs. 5-6). During this period, a total of 153 mm of rain fell (up to 120 mm on the 21$^{st}$ July 2017 alone, which was the most intense rainfall event since 1936), resulting in a 1.5 m rise in the water table. A third survey happened six days after the 21$^{st}$ July 2017 storm, with 22 mm of rain falling in one day. The material eroded from the canyons was deposited

at the base of the cliffs as gravel cones, which were remodelled by debris flows during ensuing precipitation events and subsequently eroded by wave action.





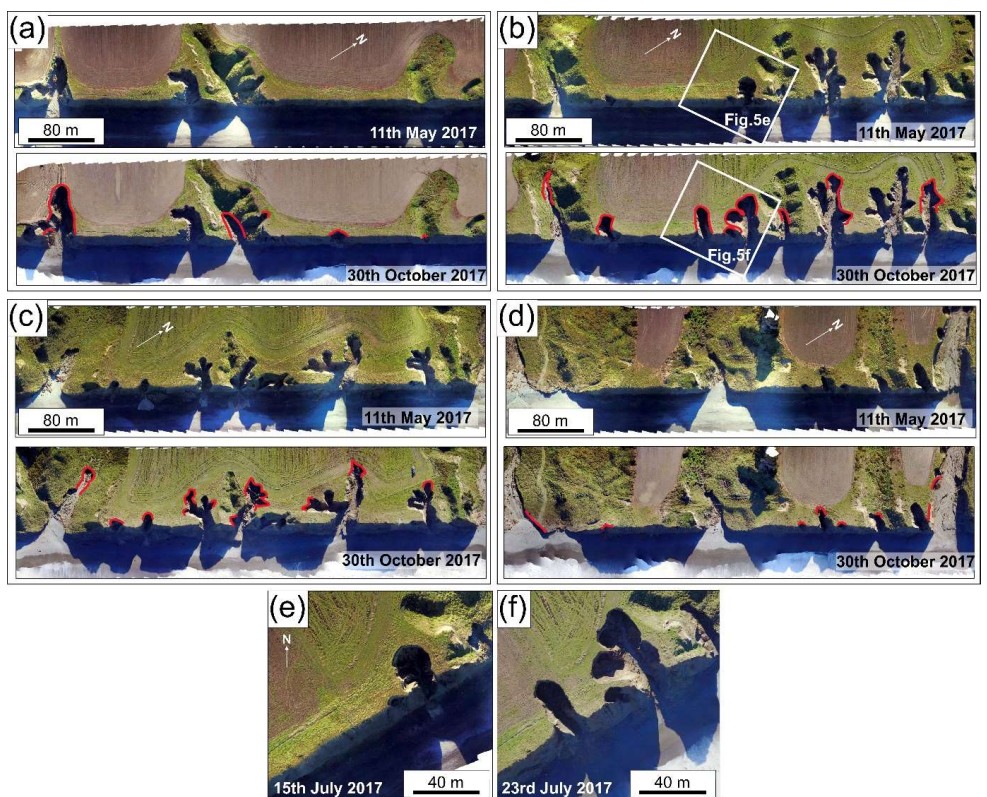

**Figure 5: (a-d) Orthophotographs of the study area at the start and end of the UAV surveys, ordered from south-west to north-east. Red lines mark eroded areas. Location in Fig. 1b. Orthophographs from a part of the study area on the (e) 15ᵗʰ July 2017 and the (f) 23ʳᵈ July 2017. Location in b.**


Earth **Surface**
**Dynamics**
Discussions

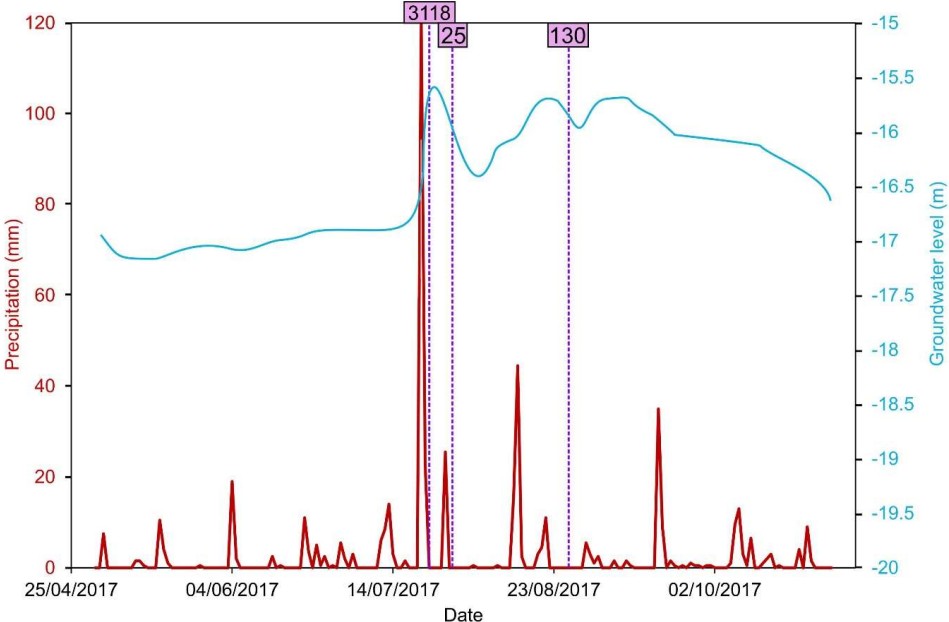

**Figure 6: Daily precipitation (for Ashburton Council) and groundwater level records (from a well located 10 km north-east of the study area) for the period 1st May to 31st October 2017 (source: Environment Canterbury). The pink lines mark the surveys when box canyon erosion was observed (the value in the pink box corresponds to the eroded area in m².).**

### 4.4.2 Long terms morphological changes

For the period 2004-2015 we have used satellite imagery to map the formation of 6 new box canyons and the elongation of 22 box canyons. 18 of these erosion episodes are recorded in the image taken on 26th August 2013 (Fig. 7a). This follows a major rainfall event between the 16th and 23rd June 2013, when 171 mm of rain fell in 7 days (with up to 51 mm falling in one day) (Fig. 7b). The other erosion episodes include the 5 box canyons eroded by the 28th March 2009, after a storm of 46 mm per day on the 31st July 2008, and the 5 box canyons eroded by the 19th October 2015, after a storm of 43 mm per day on the 19th June 2015.

Earth **Surface**
**Dynamics** Open Access
Discussions
EGU

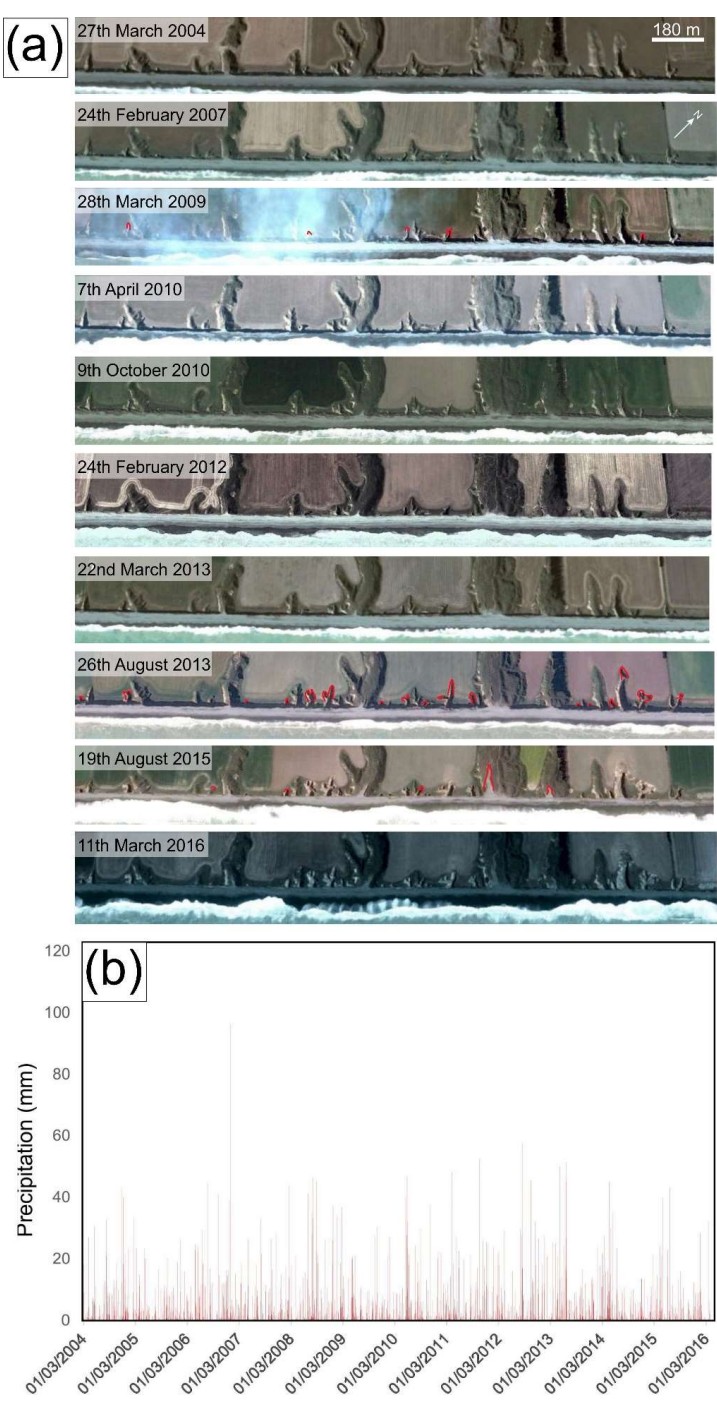

**Figure 7: (a) Satellite imagery of the study area between the 27$^{th}$ March 2004 and 11$^{th}$ March 2016 (source: Google, Maxar Technologies). Eroded areas are marked by red lines. (b) Daily precipitation record for this period for Ashburton Council (source: Environment Canterbury).**



### 4.5 Geophysical data


The location of the G-TEM transects is shown in Fig. 1b. An attempt was made to invert the G-TEM slingram-mode responses with 30-m TX-RX offset using 1-D Occam inversion. A representative inversion result is shown in Fig. 8a. The resistivity model is presented in the right panel, whereas the corresponding model-response with the actual data points is shown on the left. The best calculated smooth depth profile clearly does not fit well with

the measured signal and there is excessive structure in the ~10-20 m depth range, including the very low resistivity layer ~$10^{-4}$ $\Omega$m at depths in excess of ~12-15 m. The resistivity values between 40 and 100 m depth are lower than sea water resistivity (0.3 $\Omega$m), which is not reasonable. The inability to fit a 1-D model to the slingram responses suggests that the geoelectrical sub-surface structure is strongly heterogeneous within the footprint of the G-TEM transmitter. As a result, we cannot trust 1-D inversions of the slingram-mode data in

such a 3-D geological environment.

Earth **Surface**
**Dynamics**
Discussions

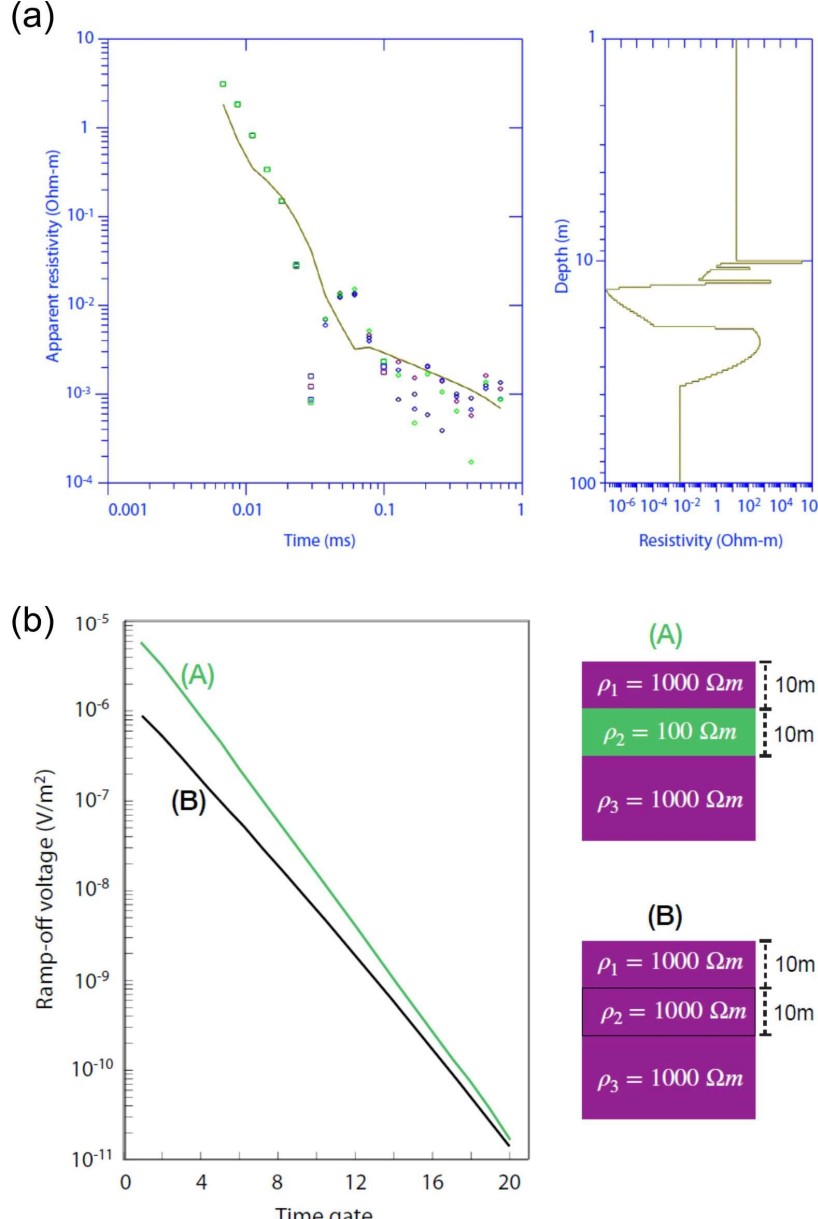

**Figure 8: (a) 1-D inversion result for data at a station located 6 m from start of Profile May15-1 (location**
**in Fig. 1b). (b) G-TEM slingram responses for model (A) containing a conductive zone at a depth of 10-20**
**m, and for model (B) without the conductive zone.**

Instead of performing 1-D inversions, we present time-gate plots for all three transects. Specifically, the amplitude of the G-TEM slingram response (in units of $10^{-10}$ V/m$^2$) at the first time-gate is plotted as a function of station number along a profile. Figure 8b displays a 1-D model (A) that contains a conductive layer of 100



Ωm between 10-20 m depths in a homogeneous 1000 Ωm background. This model generates a G-TEM slingram response that has a substantially larger amplitude at all time gates than does the model (B) without the conductive layer. Thus we regard an enhancement of response at the first time gate as indicative of a conductive zone at depth beneath the slingram station.


The first-time-gate profile of transect May15-1 is located upslope of small but recently eroded box canyons (Fig. 9a). Near the middle of this transect there is a distinctive peak that is much higher than the background. The peak is ~20-30 m wide and it appears in a similar fashion on each of the gates 1 through 7 (not shown here), although it cannot be clearly observed after gate 7. Transect May 15-2 is located upslope of recently eroded box canyons in the south-west and relatively less active canyons in the north-east of the investigated area (Fig. 9b). Lateral variations are evident along the 192 m length of the profile. The high amplitude response at the start of the profile (going from the south-west to north-east) is followed by a drop in amplitude near the midpoint of the profile, following which there is continuous fluctuation at a lower amplitude until the end of the profile. The fluctuating signals remain similar in shape for time-gates 2 to 6 (not shown). G-TEM slingram profile May17-2 was acquired upslope of the tributary of a large box canyon covered by mature vegetation (Fig. 9c). As shown in Sect. 4.4, the size and location of this box canyon have been persistent over recent years, in contrast to the neighbouring, smaller box canyons that are under active development. Transect May 17-2 shows a lower amplitude response in comparison to the previous two transects (Fig. 9c). Based on all three profiles, a general observation that can be made is that the first-time-gate amplitude of the slingram response is higher upslope of the more recently active box canyons.



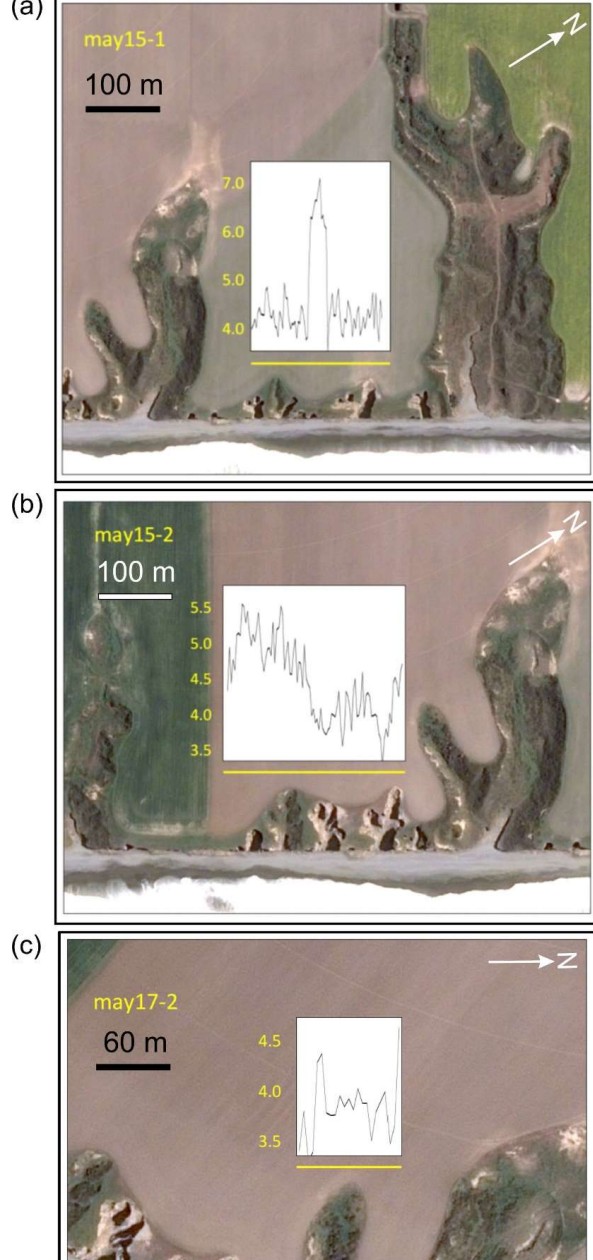

**Figure 9: First-time-gate profiles of G-TEM slingram transects (a) May15-1, (b) May 15-2, and (c) May17-2 (units in yellow are $10^{-10}$ V m$^{-2}$). Source of background imagery: Google, Maxar Technologies.**




### 4.6 Modelling

#### 4.6.1 Slope stability modelling


##### (a) Slope with sand lens

The factory of safety of the slope prior to any rainfall event was 2.514. During the first scenario ($(I-D)_3$), the factor of safety decreased to 1.371 due to undermining by tunnelling associated to high pore pressures within the sand lens, and then to 0.614 as a result of a decrease in the shear strength of the lower slope material due to an increase in pore pressure (Figs. 10a, 11a). A rainfall intensity of 40 mm per day is required to bring the factor of safety below 1 (Fig. 11c). In the case of the second scenario ($(I-D)_{14}$), changes in pore-water pressure did not result in either tunnelling or slope failure. This only resulted in a decrease in the effective stress and in the factor of safety (1.216) (Figs. 10b, 11b).




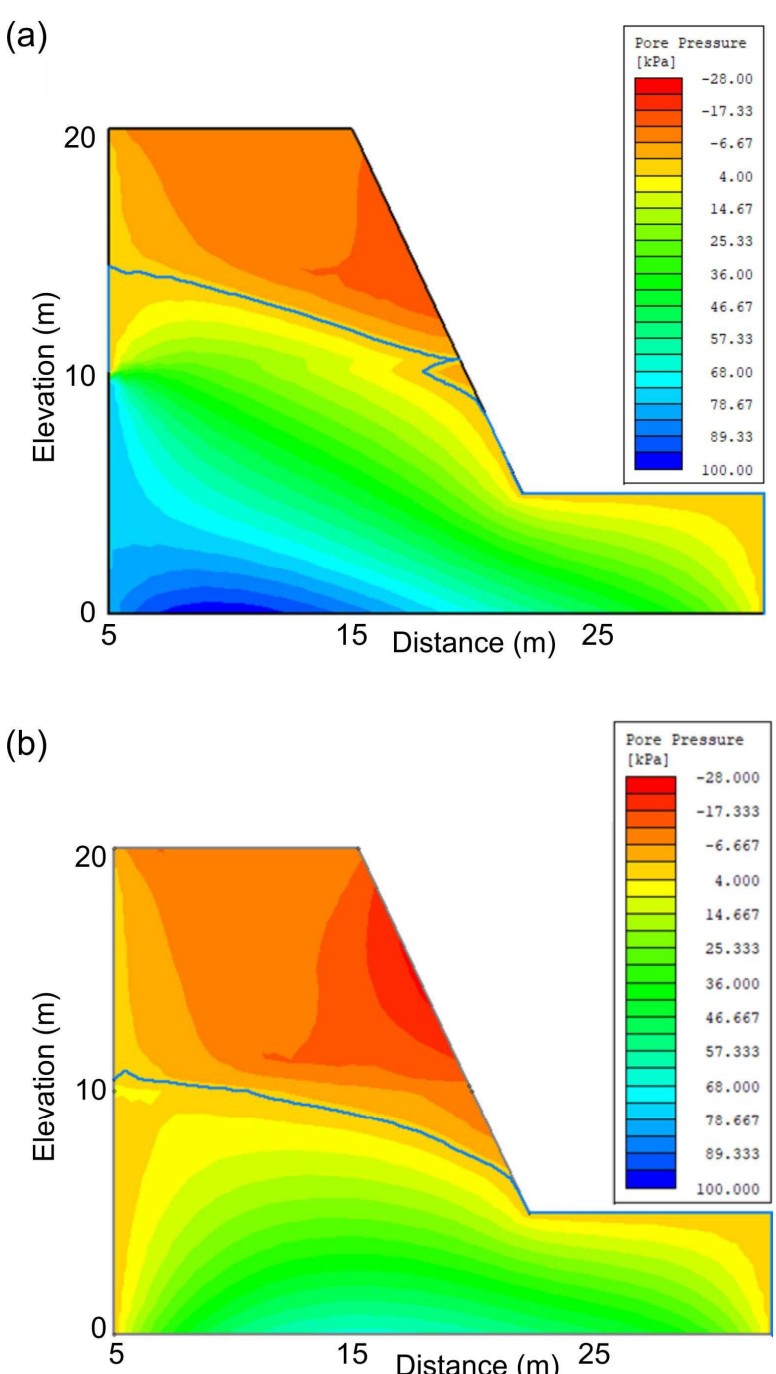

**Figure 10: Model results for sandy gravel slope with sand lens. Estimated pore water pressure after (a) 3 days for first scenario ((I-D)₃) and (b) 14 days for second scenario ((I-D)₁₄). Blue line denotes wetting front.**



Earth **Surface**
**Dynamics**
Discussions

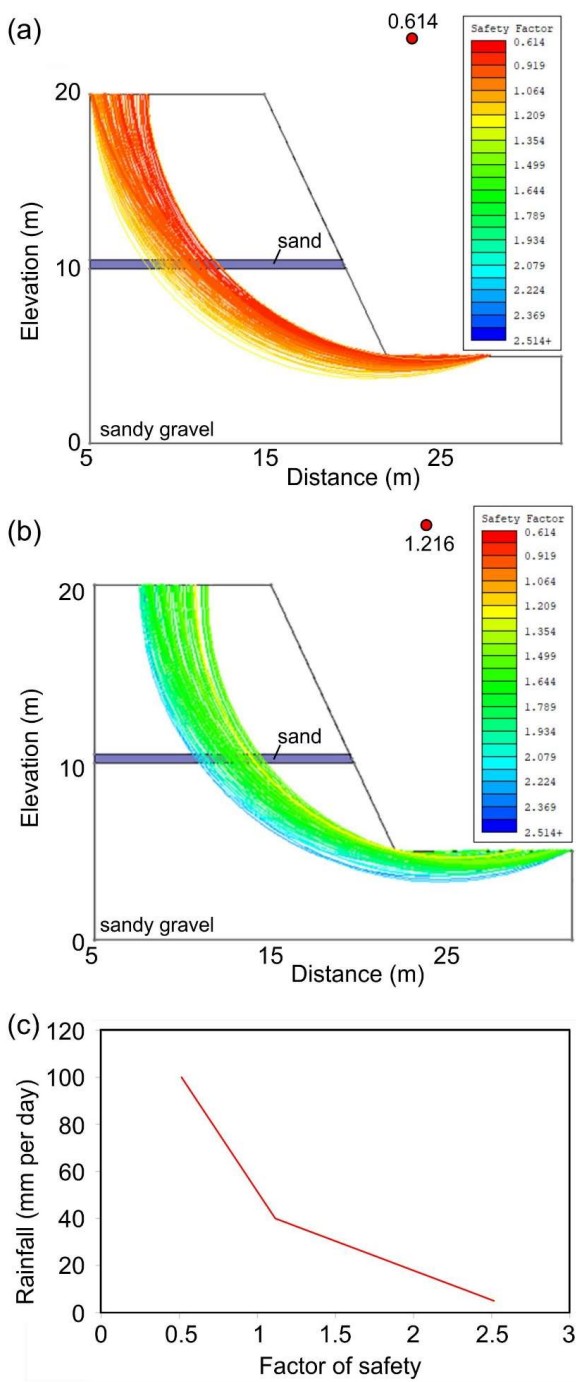


**Figure 11: Model results for sandy gravel slope with sand lens. (a) Estimated factor of safety after 3 days for first scenario ((I-D)$_3$). (b) Estimated factor of safety after 14 days for second scenario ((I-D)$_{14}$). (c) Plot of rainfall intensity vs. factor of safety for the for first scenario ((I-D)$_3$ for the slope with sand lens.**





*(b) Slope with gravel lens*

The factor of safety of the slope prior to any rainfall event is 1.793. For the first scenario ((I-D)$_3$), the factor of safety decreased to 1.166, and neither tunnelling nor slope failure occurred (Figs. 12a-b). In the case of the second scenario ((I-D)$_{14}$), the outcome is the same, with the factor of safety decreasing to just 1.588 (Figs. 12c-d).

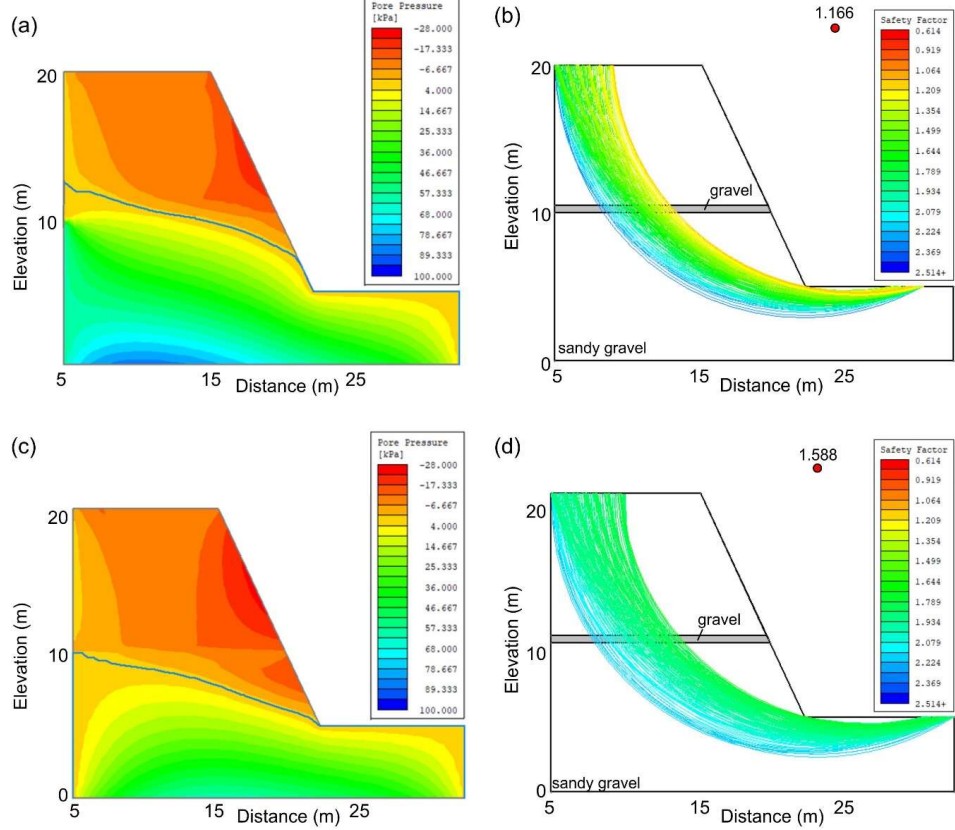

**Figure 12: Model results for sandy gravel slope with gravel lens. (a) Estimated pore water pressure after 3 days for first scenario ((I-D)$_3$). (b) Estimated factor of safety after 3 days for first scenario ((I-D)$_3$). (c) Estimated pore water pressure after 14 days for second scenario ((I-D)$_{14}$). (d) Estimated factor of safety after 14 days for second scenario ((I-D)$_{14}$). Blue line in a and c denotes wetting front.**

### 4.6.2    LEM

The model results for the two erosional modes and two rainfall scenarios are displayed in Figs. 13-16 and discussed in more detail below.



The results of the stream power law model with a high intensity rainfall scenario are shown in Fig. 13. During the first hour and as a result of the low recharge value ($1 \times 10^{-9}$ m s$^{-1}$), the water table drops slowly and no erosion

occurs. All parameters, except for shear stress, decrease rapidly. An intense storm (recharge of $2.8 \times 10^{-5}$ m s$^{-1}$) starts after 1 hr; groundwater seeps out at the cliff base, where the shear stress is highest, forming a gully between the cliff base and the beach. In the ensuing 2 hr, the recharge rate stays the same but the water table height in the sandy area decreases due to the high permeability; as a result the shear stress decreases and erosion stops. At the third hour, the storm stops and there is no recharge; the water table drops slowly and the shear stress and

groundwater seepage decrease. The results of the stream power law model with a low intensity rainfall scenario are similar to those of the high intensity rainfall scenario (Fig. 14).




Earth **Surface**
**Dynamics**
Discussions

EGU

**Figure 13: Evolution of the main simulated parameters for the stream power law with high intensity rainfall scenario. Section A shows the evolution curves of the maximum value of the main parameters. Section B shows the topographic evolution during the first 4 hours. White arrow indicates the gully.**




**Figure 14: Evolution of the main simulated parameters for the stream power law with low intensity rainfall scenario. Section A shows the evolution curves of the maximum value of the main parameters. Section B**

**shows the topographic evolution during the first 4 hours. White arrow indicates the gully.**

The results of the linear diffusion model with high intensity rainfall scenario are shown in Fig. 15. During the first hour of the simulation, the recharge is low ($1 \times 10^{-9}$ m s$^{-1}$); the water table height, shear stress, diffusivity coefficient and groundwater flux decrease very slowly, whereas the surface water discharge increases. After the





first hour, an intense storm increases the recharge to $2.8 \times 10^{-5}$ m s$^{-1}$. The water table height rises quickly (from 5.9 to 6.5 m), as do all the parameters, reaching their maximum values. A small box canyon develops at the base of the cliff where the shear stress is highest. The intense storm continues for the next 2 hours, and the water table height rises quickly to its maximum value of 7.1 m. The main parameters decrease, and the box canyon enlarges to 16.5 m × 11.3 m, eroding into the cliff. At the end of the storm there is no recharge, the water table drops

slowly, the shear stress and the groundwater seepage decrease rapidly, and erosion stops. During the rest of the simulation there is no recharge; the water table continues to drop, the shear stress and the groundwater seepage decrease, and no erosion occurs.

Earth **Surface**
**Dynamics**
Discussions



**Figure 15: Evolution of the main simulated parameters for the linear diffusion with high intensity rainfall scenario. Section A shows the plots of the maximum value of the main parameters. Section B shows the topographic evolution during the first 4 hours. White arrow denotes the direction of erosion.**

The linear diffusion with low intensity rainfall model (Fig. 16) starts with a low recharge ($1 \times 10^{-9}$ m s$^{-1}$) during the first hour. During this period, the water table drops slowly, the main parameters (shear stress, diffusion coefficient groundwater flux and surface water discharge) decrease rapidly and there is no erosion. After 1 hr, a low intensity storm starts (recharge of $1.12 \times 10^{-5}$ m s$^{-1}$). The water table height increases slowly; shear stress, groundwater flux



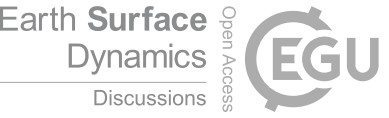

and surface water discharge increase, whereas diffusion continues to decrease gradually. The erosion process forms a scarp at the base of the cliff. The storm continues for 4 hr. The water table rises slowly, but the high

permeability in the sand unit does not allow the water table to increase in this area. All parameters continue to increase, but their values are not high enough to erode the cliff further. After 6 hr, the storm stops; there is no recharge, the water table drops slowly, and the values of the main parameters decrease.



Earth **Surface**
Dynamics
Discussions

**Figure 16: Evolution of the main simulated parameters for the linear diffusion with low intensity rainfall scenario. Section A shows the evolution curves of the maximum value of the main parameters. Section B shows the topographic evolution during the first 4 hours. White arrow marks the scarp.**

Finally, the results of the stream power law model without groundwater seepage show a set of closely-spaced gullies, growing from the base of the cliff upwards. The largest of these are incised into the sand unit (Fig. 17).





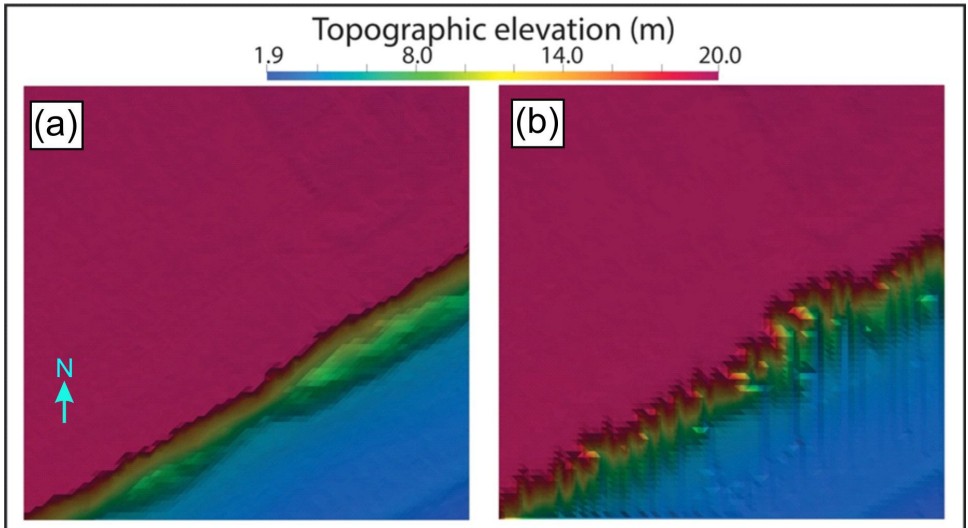

**Figure 17: Topographic evolution using the stream power law without groundwater seepage. (a) Initial topographic conditions. (b) Topographic conditions after 24 hours.**

**5       Discussion**

Box canyons are characteristic landforms along the Canterbury coast (Fig. 1a). They are an important driver of coastal geomorphic change as well as loss of agricultural land. The morphologic attributes of the box canyons are similar to those associated with groundwater seepage elsewhere (Table 1).


The Canterbury box canyons initiate and evolve via two types of groundwater-related processes. The first process is seepage erosion of sand, which leads to the formation of alcoves and tunnels (Fig. 4). This lowers the overall factor of safety of the slope and is a precursor to the second process, which is slope failure. Box canyons primarily evolve by retrogressive slope failure (Figs. 5, 7), which results in the elongation of the box canyons and, to a lesser

extent, in widening and branching along the canyon walls. The shear stress of the water seeping out of the canyon head and flowing along the base of the canyon is high enough to entrain the failed sandy material (shear stress of $7.5 \times 10^{-5}$ in Fig. 15a; Petit et al. (2015)), which explains why the floors of box canyons are primarily covered by gravel. Wave erosion removes the failed material at the canyon mouths and the base of the cliff. Isotropic scaling of length with width (Fig. 3c) suggests that canyon planform shape is generally geometrically similar at

consecutive stages of evolution.

The field observations of box canyon morphology and development are most similar to the LEM results for the high intensity rainfall scenario of the linear diffusive erosion model (Fig. 16). There are two reasons for this: first, as observed in Culling (1963) and Barnhardt et al. (2019), the retrogressive slope failure is best simulated by a

diffusive linear model; secondly, our linear diffusive erosion model links the diffusion co-efficient with the shear stress generated by the groundwater. The evolution of the different models and scenarios appears to be influenced by the initial topography - with the cliff preventing the upslope development of erosive features in the stream



power law mode, regardless of the recharge values – and the water table height, which depends on permeability and recharge (Figs. 13-16). In the stream power law mode, erosion is linked to surface water, which depends

exclusively on groundwater seepage. Since the water table can only intersect the beach or the cliff surface, surface erosion of the plain above the cliff top is not possible (Figs. 13-14). When the stream power law model does not depend on groundwater seepage, the surface erosion does affect the cliff face, but it results in a series of closely spaced and narrow gullies (Fig. 17). The conceptual model, the program configuration and the initial and boundary conditions make the LEM outcomes only applicable to box canyon formation at this site.


The location of the box canyons is controlled by two factors. The first factor is the occurrence of sand lenses across a sandy gravel cliff face. This geological framework is conducive to alcove formation, tunnelling and slope failure. The higher permeability of the sand and clean gravel lenses, in comparison to the surrounding sandy gravel (Table 2), facilitates faster water transfer to the cliff face, as confirmed by the LEM. Alcoves and tunnels only

form in the sand lenses, however, because the latter develop higher pore pressures, and sand is easier to entrain and remove in comparison to clean gravel in view of its lower shear strength. Slope failure only occurs in sandy gravel slope with sand lenses. The higher pore pressure developed in the sand lenses is transferred to the sandy gravel slopes, resulting in a larger decrease in the shear strength and higher water table in comparison to the sandy gravel slope with gravel lens (Figs. 10-12). The second factor is a hydraulically-conductive zone upslope of the

box canyon. This inference is supported by the following observations: (i) Braided river channel infills, which tend to comprise highly permeable, coarse grained materials (Moreton et al., 2002), lead into the box canyons heads (Figs. 3a-b); (ii) Clustered distribution of box canyons between the two braided rivers with the highest flow rates (Rakaia and Rangitata Rivers (Environment Canterbury, 2019)) (Fig. 1a); (iii) Geophysical observations (Fig. 9). We interpret the higher amplitude response in G-TEM profiles located upslope of recently active box

canyons as buried groundwater conduits made up of gravel and/or sandy units (e.g. Weymer et al., in review), or tunnels formed by sub-surface water flow in sand units. Further analysis of the G-TEM data, including 2-D modelling and inversion, is required to ascertain the sub-surface hydraulic geometry responsible for the along-profile amplitude variations. The above confirm the importance of spatial variations in hydrogeological properties as a factor controlling the location of a box canyon. This had initially been suggested by Dunne (1990) and has

been documented for box canyons in bedrock environments (Laity and Malin, 1985; Newell, 1970). Development of box canyons downslope of permeable conduits may also explain why most of the erosion entails elongation of existing canyons, rather than formation of new ones (Figs. 5, 7). It also agrees with the results of experimental modelling by Berhanu et al. (2012), which suggest that channels grow preferentially at their tip when the groundwater flow is driven by an upstream flow.


Box canyon formation is rapid (daily timescales) and recent (<3 years ago). It is an episodic process that occurs after a threshold is exceeded. This threshold entails a rainfall intensity of >40 mm/day (Figs. 6, 7b, 11, 15), which occurs once every 227 days, on average (Fig. 7b). According to our LEM, up to 100 m³ of water are estimated to have seeped out of the cliff face to erode 2540 m³ of material (Fig. 15), which contrasts with the inference by

Howard (1988) that 100-1000 times more water than volume of eroded sediment must be discharged in order to create a sapping valley. The erosion rate documented in our study area is up to 30 m per day (Figs. 5e-f), which is the highest rate documented for box canyons so far.



The majority of the box canyons in our study area have shown evidence of erosion in the past 11 years. The OSL
data, however, suggest that two largest box canyons have largely been inactive during at least the last 2 ka (recent
erosion is only documented in the central section of their mouths, Figs. 5-7). This contrasts with the inference by
Schumm and Phillips (1986) that they were formed by spillage of water from swamps behind the cliffs in the 19$^{th}$
century. Instead we propose that the largest box canyons are relict features that formed as a result of higher
groundwater flow, and possibly surface erosion, in the past. The age of sample NZ13A suggests that this may
have occurred during the Last Glacial Maximum.

## 6    Conclusions

Box canyon formation is a prevalent process shaping the Canterbury coast of the South Island of New Zealand.
In this study we have integrated field observations, OSL dating, multi-temporal UAV and satellite data, time-
domain electromagnetic surveying, and slope stability and landscape evolution modelling to constrain the
controlling factors and temporal scales of box canyon formation. Our results indicate that box canyon development
in sandy to gravel cliffs is a groundwater-related, episodic process that occurs when rain falls at intensities of >40
mm per day. At the study area, such rainfall events occur at a mean frequency of once every 227 days. Box
canyons have been developing, primarily by elongation, in the last 11 years, with the latest episode dating to 3
years ago. Canyons form within days and erosion rates can reach values of up to 30 m per day. The two largest
box canyons in the study area, however, have not been actively elongating or widening in the last 2 ka. The key
processes responsible for canyon development are the formation of alcoves and tunnels in sandy lenses by
groundwater seepage erosion, followed by retrogressive slope failure. The latter is a result of undermining and a
decrease in shear strength due to excess pore pressure development in the lower part of the slope. The location of
the box canyons is controlled by the occurrence of hydraulically-conductive zones, which comprise relict braided
river channels and possibly tunnels, and sand lenses exposed across the sandy gravel cliff. Box canyon formation
is best represented by a linear diffusive model and geometrical scaling.

## 7    Code and data availability

We used Drone Deploy (https://www.dronedeploy.com/), IXG-TEM (http://www.interpex.com/), Slide2
(https://www.rocscience.com/software/slide2) and Landlab (https://landlab.github.io/) in this paper. Only the
latter is freely available. All data from this study appear in the tables, figures, main text and supplementary
materials.

## 8    Author contribution

A.M. designed the study and drafted the manuscript, which was revised by all co-authors. A.M., R.M, P.P., M.E.
and B.A.W. participated in the fieldwork. R.M. and R.P.T. interpreted the UAV data and satellite imagery. N.S.,
R.C.G. and D.C. carried out the slope stability and landscape evolution modelling. P.P. and M.E. processed the
geophysical data. A.A. and A.T.G. were in charge of the optically stimulated luminescence dating.



## 9 Competing interests


The authors declare that they have no conflict of interest.

## 10 Acknowledgments

We are grateful to Robbie Bennett, Clark Fenton, Daniele Spatola and Philippe Wernette for their assistance during fieldwork, and to Environment Canterbury for the provision of data.

## 11 Financial support

This project has received funding from the European Research Council (ERC) under the European Union's Horizon 2020 research and innovation programme (grant agreements No 677898 (MARCAN) and No 678106 (INTERTRAP)).

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
