# Peer review of "Box canyon erosion along the Canterbury coast (New Zealand): A rapid and episodic process controlled by rainfall intensity and substrate variability"

_Earth Surface Dynamics, 2020_

## Referee Comment (RC1) · Anonymous Referee #1 · 1 Jul 2020

Micallef et al present an intensive study of the erosion of the Canterbury Coastline of New Zealand. The authors identity what they call "box canyons" along the coastline and investigate what may be the controlling factors driving this erosion. Their methods range from geophysical surveys, landscape evolution and slope stability modeling, mapping and aerial photography, and luminescence geochronology. The authors conclude that the emission of groundwater at the surface and the resulting slope instability from the increase in pore pressure drives failure and subsequent erosion and that this can be modeled with a linearly diffusing landscape evolution model.

[Figure]

I think this paper should be rejected to give the authors plenty of time to reorganize the paper and reperform the science to address the problems in the paper. Here is why:

1) The Landscape evolution modeling, on which the authors' interpretation's rely on heavily, is wrong. Details are given further on in this review.

Because the evidence provided from the landscape evolution modeling is not robust, I fear the authors made need to redo their entire LEM analysis including reperforming model runs, including calibrating their parameter values and conducting sensitivity analyses. I fear that the authors will need more time than is offered during Esurf's review process. As of now, I find the authors statements in the discussion unconvincing.

2) The data presented are not fully synthesized into a clear and complete story or analysis. The challenge here is that it is hard to understand how each of the data the authors present ties into the overall picture of 'box canyon' erosion. As such, it is not clear as the paper is currently written what the contribution of the paper is towards understanding Earth surface dynamics. After reading the paper, the only things I got out of it are that the authors did a lot of work collecting a lot of data, but I can't say I understand what the contribution is. Two good examples are the luminescence ages and the geophysical resistivity surveying. The ages are used mostly in a passing way and the geophysical resistivity doesn't seem like it was very successful, so it not clear why it is included?

3) The term Box Canyon doesn't bring to mind the features that the study focuses on. I understand box canyons to be canyons with vertical walls, flat bottoms, and generally ingress and egress is only possible through the mouth of the canyon. The classic box canyon are the narrow canyons typically associated with the arid western United States. I think what the authors are studying are actually gullies that progress into canyons when they collect enough drainage area. As the authors note in Lines 383-385, the younger and lower-discharge/drainage areas/stream orders of these features have V and U shaped cross sectional profiles. Whereas the typical box canyons I have
seen in the field continue to have flat bottoms and vertical walls even at low stream orders.

I think this will be a problem for the authors in that researchers looking for information on box canyons will not consider this work relevant, and those studying coastal gully erosion will miss this paper due to the title and mislabeling of the features of interest. Instead, I think the authors should rewrite this paper while avoiding the use of the term box canyon. Either referring to these as coastal cliff gullies, headward cliff erosion, or some new term the author's come up with.

4) The details and protocol on the luminescence age laboratory methods are great; the information on what was dated and why is not.

After reading the paper, I could not tell what landform the authors dated using luminescence. This is pretty important for interpreting the ages.

As I do not have expertise in slope stability modeling or geophysical resistivity surveys, I am not going to comment on them.

Landscape evolution modeling:

The authors use two models to simulate the erosion of the Canterbury Coast. The first is the stream power model and the second is a linear diffusion model. Neither of these is likely appropriate to simulate the erosion here, nor do the authors provide the necessary analysis to justify their use. First, the stream power model is really for detachment-limited landscapes. However, the authors do not make a case that the erosion and sediment transport conditions here justify the use of the stream power model. Particularly, as the presence of major slope failures and alluvium in the channels suggests that a significant sediment cover effect may be happening here.

Next, the authors use a linear diffusion model as another endmember. However, with the presence of major slope failures, which are significantly non-local and thus not diffusive, it is hard to justify its use and the authors do not provide this justification in
the manuscript. The discussion section completely disregards this fact and overall is not convincing.

Another problem in the methodology occurs with the values used in the model. One problem is when the authors equate the values for K and D in the stream power model and the linear diffusion model. These values represent very physically processes and are likely very different, by orders of magnitude in some cases. Their use of also assuming that K/D is proportional to the seepage flux and surface water shear stress, while I am open to the idea, isn't backed up with a reference or proof of concept. Next the authors state that they obtain the values for M and Tau_t "by trial and error." What does this mean? Did they just pick values until they obtained behavior they wanted? Because the the authors picked values by trial and error, rather than by constraining them from some means, it is not clear if the modeling results presented are meaningful. Normally, when one does not know what input parameters should be, a sensitivity analysis is needed to evaluate the outcomes from a full range of possible values for a specific parameter.

Finally, there is no sensitivity analysis given on how model outputs change with change in the input parameters. This is needed to interpret the results from landscape evolution modeling. On line 297, it says that the authors are testing the models, but these methods are not really a formal test of these LEMs as currently written.

One line 337: Actually, there probably is a way in landlab to incorporate sand lenses, but this would essentially involve developing a new landlab component to do so.

Why is Equation 10 is the same as Equation 7?

Luminescence Dating I think that the laboratory methods in the supplemental is generally well done. I like the level of detail and the well-documented dose recovery and fading tests. It seems like these might have been challenging samples.

Dose recovery tests:

These values don't seem all that great to me. I generally try changing my protocol if the dose recovery ratio comes out worse than 1.10 or 0.9. What could be causes of this not-great dose recovery? Is this typical for feldspar from New Zealand?

Those g-values might not be insignificant. If you perform the fading correction on the k-feldspar for NZ14A, do the ages agree with the polymineral?

Perhaps the discrepancy in age between the polymineral fine grain and the k-feldspar (for NZ14A) is due to the presence of unknown minerals in the polymineral aliquots? It is possible that there are minerals in there with unknown pIRIR properties that might be transferring charge in unexpected ways?

One other thing that might be useful is if the authors could report the distribution of the residual dose. If there is large variance, maybe this could explain the discrepancy of the polymineral and feldspar ages.

One fairly big issue is that there is limited reporting of what landform the luminescence samples were collected from. The locations are given on the aerial photo, but I cannot determine if they were collected from sediment in the channel or from the walls of the canyon, etc. Because of this, it is hard to interpret the interpretations of partial bleaching fully. As the pIR290 ages overestimate the pIR225 ages, there is likely to be partial bleaching, but how much? Are the ages themselves suspect or not? How do you know? Do you need a lower temperature pIR signal to get a better bleached age?

---

## Short Comment (SC1) · 9 Jul 2020

The reviewer states that "the geophysical resistivity doesn't seem like it was very successful, so it not clear why it is included?"

The inability to fit a 1-D resistivity-vs-depth model to the observed slingram responses, coupled with the observation that the first-time-gate response profiles show considerable along-transect variability (Figs. 9a-c), suggests that the geo-electrical subsurface structure is strongly heterogeneous within the footprint of the G-TEM transmitter.

[Figure]

The geo-electrical heterogeneity suggested by the G-TEM responses could be caused by the resistivity signature of discrete subsurface groundwater conduits, perhaps in the form of cylindrical tubes of irregular cross-section, carrying groundwater seaward across the Canterbury Plains and discharging either at the cliffs or the beach.

This is the key information provided by G-TEM in support of the hypothesis of discrete bodies of groundwater flowing along preferential seepage pathways and supporting the growth and evolution of box canyons along the coastal cliffs.

Work is currently underway, as part of a different study, on 2-D forward modelling of the slingram responses; our preliminary results indicate that the G-TEM data can be explained by the aforementioned electrically conductive groundwater conduits. It suffices to say here that, to depths of several tens of metres, the subsurface landward of the box canyons is certainly not one-dimensional from the perspective of electrical resistivity.

---

## Author Comment (AC1) · 9 Jul 2020

Here I address two general issues raised by reviewer 1. The issues related to landscape evolution modelling, geophysical data, and OSL measurements are addressed by the responsible co-author as personal short comments (Roger Clavera-Gispert, Mark Everett and Alida Timar-Gabor, respectively).

"2) The data presented are not fully synthesized into a clear and complete story or analysis. The challenge here is that it is hard to understand how each of the data the

authors present ties into the overall picture of 'box canyon' erosion. As such, it is not clear as the paper is currently written what the contribution of the paper is towards understanding Earth surface dynamics. After reading the paper, the only things I got out of it are that the authors did a lot of work collecting a lot of data, but I can't say I understand what the contribution is. Two good examples are the luminescence ages and the geophysical resistivity surveying. The ages are used mostly in a passing way and the geophysical resistivity doesn't seem like it was very successful, so it not clear why it is included?"

In the revised version of the manuscript we will sub-divide and update the Discussion by clearly highlighting what inference each data set allows us to make, and how these can be integrated to reach the conclusions that we present at the end.

In our contribution we address two points: (i) the role that lithology and permeability play in box canyon initiation and evolution, and (ii) the temporal scale of box canyon formation. Because of the reliance on experiments and numerical models, and the paucity of process-based observations and instrumental analyses, previous studies had not thoroughly addressed these issues in the past. By integrating field data with modelling, we show that box canyon formation is an episodic process associated to groundwater flow that occurs when a threshold rainfall intensity is exceeded. The canyons in our study area are actively eroding, and erosion rates can be as high as 30 m per day. Hydraulically-conductive zones (e.g. relict braided rivers channels, tunnels) and sand lenses exposed at the cliff face control the location of the box canyons. We believe that these outcomes are important because: (i) there are only a few places where the mechanisms by which seepage erosion occurs have been clearly defined, and (ii) the results can be used to test and quantify models of canyon formation, and improve the reconstruction/prediction of landscape evolution by groundwater-related processes.

The points related to OSL data and geophysics are addressed in separate comments by Alida Timar-Gabor and Mark Everett.

**ESurfD**
[Figure]

"3) The term Box Canyon doesn't bring to mind the features that the study focuses on. I understand box canyons to be canyons with vertical walls, flat bottoms, and generally ingress and egress is only possible through the mouth of the canyon. The classic box canyon are the narrow canyons typically associated with the arid western United States. I think what the authors are studying are actually gullies that progress into canyons when they collect enough drainage area. As the authors note in Lines 383-385, the younger and lower-discharge/drainage areas/stream orders of these features have V and U shaped cross sectional profiles. Whereas the typical box canyons I have seen in the field continue to have flat bottoms and vertical walls even at low stream orders. I think this will be a problem for the authors in that researchers looking for information on box canyons will not consider this work relevant, and those studying coastal gully erosion will miss this paper due to the title and mislabeling of the features of interest. Instead, I think the authors should rewrite this paper while avoiding the use of the term box canyon. Either referring to these as coastal cliff gullies, headward cliff erosion, or some new term the author's come up with."

The reviewer's point is valid. We will refer to these landforms as coastal gullies.

---

## Referee Comment (RC2) · Anonymous Referee #2 · 13 Jul 2020

[referee-annotated manuscript omitted]

---

## Short Comment (SC2) · 13 Jul 2020

*Reviewer: These values don't seem all that great to me. I generally try changing my protocol if the dose recovery ratio comes out worse than 1.10 or 0.9. What could be causes of this not-great dose recovery? Is this typical for feldspar from New Zealand?*

**Answer**: The authors would like to thank the reviewer for the keen observations. So far, several studies reported poor dose recovery results for pIRIR$_{290}$ (e.g. Stevens et al., 2011; Thiel et al., 2011, 2014; Lowick et al., 2012; Roberts et al., 2012; Murray et al., 2014, Veres et al., 2019; Avram et al., 2020). Usually, these poor dose recovery ratios were greater than unity. On the other hand, Sohbati et al., 2016 reported good dose recovery ratios for K-feldspars using pIRIR$_{290}$ extracted from palaeorockfall boulders from South Island, New Zealand.

Qin and Zhou (2012) suggested that the dose recovery ratio is dependent on the test dose size. Yi et al., 2016 concluded that the best dose recovery ratios are found when the test dose ranges between 15 and 80% of the total dose. Here, following this suggestion dose recovery measurements were carried out using a test dose ranging between 20 and 60% of the total dose. On the other hand, a recent study by Avram et al. (2020) reported poor dose recovery ratio using a test dose of 41% of the total dose on polymineral fine grains extracted from Serbian loess.

As the reviewer suggested, two protocols are actually being used, namely pIRIR$_{290}$ and pIRIR$_{225}$. As presented in the manuscript "the ratios between the recovered and given dose for polymineral fine grains are 1.11±0.03 (NZ13A) and 1.01±0.03 (NZ14A) using piRIR$_{225}$ protocol and 1.14±0.05 (NZ13A) and 1.03±0.06 (NZ14A) using pIRIR$_{290}$ protocol. In the case of coarse K-feldspars, the dose recovery ratios are 0.96±0.01 (NZ13A) and 0.87±0.03 (NZ14A) using pIRIR$_{225}$ protocol, while those obtained by applying pIRIR$_{290}$ protocol are 1.17±0.06 (NZ13A) and 1.16±0.04 (NZ14A)." Please note that in the case of the pIRIR$_{225}$ protocol the results are generally satisfactory – they are within 1σ uncertainty and they are consistent with the 0.9-1.1 interval. We have stated that "We conclude that the overall behaviour of the pIRIR$_{225}$ protocol is satisfactory whereas the pIRIR$_{290}$ protocol overestimates the given dose by ~ 17%."

Some authors (e.g Stevens et al., 2011; Alexanderson and Murray, 2012) concluded that poor dose recovery ratios can be the result of the incorrect residual dose estimation.

In order to avoid potential complications related to laboratory bleaching of the natural samples, the dose recovery test could be performed by adding a beta dose on top of the natural dose. Then, the dose recovery ratios are obtained by dividing the measured dose to the sum of the natural and the given dose.

Such a dose recovery test was performed in the case of pIRIR$_{290}$ protocol by adding a beta dose of 100 Gy on top of natural dose. The results are presented in the last column of the table below and compared with the previous reported values stated above and presented in the manuscript (column 3).

| Sample code | Grain size | Recovered/given ratio (residual dose subtraction) | Recovered/given ratio (100 Gy on top of natural) |
|---|---|---|---|
| NZ13A | 4-11 μm polymineral | 1.14±0.05 | 1.12±0.05 |
| NZ14A | 4-11 μm polymineral | 1.03±0.06 | 1.15±0.03 |
| NZ13A | 63-90 μm K-feldspar | 1.17±0.06 | 1.03±0.03 |
| NZ14A | 63-90 μm K-feldspar | 1.16±0.04 | 1.02±0.02 |

For coarse K-feldspars the dose recovery ratio improved from 1.17±0.06 to 1.03±0.03 for sample NZ13A and from 1.16±0.04 to 1.02±0.02 for sample NZ14A. In the case of polymineral fine grains, the dose recovery ratio for sample NZ13A was still slightly overestimated (1.12±0.05), while for sample NZ14A, the dose recovery ratio increased from 1.03±0.06 to 1.15±0.03. As such there is no clear trend. We can only conclude that while the results are not ideal, they are acceptable, and the used protocols are providing the best attainable results for these samples given state-of-the-art available methods (see also answers below regarding changing the preheat temperature).

***Reviewer: Those g-values might not be insignificant. If you perform the fading correction on the k-feldspar for NZ14A, do the ages agree with the polymineral?***

**Answer:** The g-values are not significant in our view. We have discussed this in detail in Avram et al. (2020), where we are presenting more extended fading measurements including on calibration quartz. Consequently, as we do not consider these short-term measurements to be reliable and we do not consider the values to be significant, we are not presenting corrected ages. For the sake of the exercise, following the reviewer's suggestion we have calculated the corrected K-feldspars ages ($pIRIR_{225}$) for both samples NZ13A and NZ14A. The results are presented in the next table:

| Sample code | g-value (%/decade) | Uncorrected Age (ka) | | | | | Corrected Age (ka) |
|---|---|---|---|---|---|---|---|
| | K-feldspar ($pIRIR_{225}$) | pfg ($pIRIR_{225}$) | K-feldspar ($pIRIR_{225}$) | pfg ($pIRIR_{290}$) | K-feldspar ($pIRIR_{290}$) | K-feldspar ($pIRIR_{225}$) |
| NZ 13A | 0.60±0.51 | 16±0.1 | 20.1±1.5 | 20.9±2.0 | 26.2±2.1 | 20.8±1.5 |
| NZ 14A | 0.85±0.09 | 4.6±0.4 | 1.9±0.1 | 6.0±0.7 | 3.1±0.3 | 2.0±0.1 |

The corrected pIRIR ages were calculated using the R Luminescence-package (Dietze et al., 2013) according to Huntley and Lamothe (2001) method.

As can be seen, there is no significant difference in the ages obtained.

***Reviewer: Perhaps the discrepancy in age between the polymineral fine grain and the k-feldspar (for NZ14A) is due to the presence of unknown minerals in the polymineral aliquots? It is possible that there are minerals in there with unknown pIRIR properties that might be transferring charge in unexpected ways?***

**Answer:** This is indeed a reasonable assumption that is worth further study. Indeed, the polymineral fraction consists of a mixture of different kinds of minerals but it is assumed that the measured emission in the violet-blue band is predominantly caused by feldspar minerals

(Tsukamoto et al., 2012; Kreutzer et al., 2014). Li and Wintle (1992) investigated the stability of luminescence signal of polymineral fine grains in comparison with those from K- and Na-rich feldspars, all extracted from loess. They found that the thermal stability of the IRSL signals from polymineral fine grains extracted from different areas around the world is similar, and the IRSL signal from fine grains is less stable than the signal of K-feldspars. Tsukamoto et al., 2012, found that IRSL and pIRIR signals from polymineral fine grains are less stable than K-feldspar sample when a lower preheat temperature (260–300 °C) is used, but the stability among different types of feldspars becomes similar when a higher preheat temperature (>320 °C) is included. From a dating point of view, they suggested that the IRSL in polymineral fine grains behaves similarly to the Na-feldspars signal, rather than to the K-feldspar's signal. To our knowledge there are few, if any, published luminescence dating studies that documented both polymineral and K-feldspars ages obtained by using pIRIR methods. Rahimzadeh et al., (2019 - conference abstract) reported polymineral fine grains and K-feldspars ages using pIRIR$_{225}$ protocol on Bavarian loess. They have found that the fading corrected ages for the two different grain sizes agree within errors for most of the samples, observing a larger discrepancy for samples with an age above 100 ka. These kinds of studies are worth pursuing, however; in our view it would be better to attempt such studies on sites where independent age control or at least well-behaved quartz chronologies are available.

Here, we have further investigated the growth curves of polymineral fine grains and K-feldspars (see Figures below). As can be seen, the growth curves overlap over the dose range investigated.

[Figure]

*Reviewer: One other thing that might be useful is if the authors could report the distribution of the residual dose. If there is large variance, maybe this could explain the discrepancy of the polymineral and feldspar ages.*

**Answer:** Residual doses measured for each individual aliquot are reported in the table below. There is no significant scatter in between the measured residual doses for each aliquot, and certainly not a difference that could account for the reported age discrepancy. Therefore, we conclude that this scenario could not explain the observed discrepancy between polymineral and K-feldspars ages.

| Sample code | Grain size | protocol | Residual dose | Average residual dose |
|---|---|---|---|---|
| NZ 13A | 4-11 µm polymineral | pIRIR$_{225}$ | 4.2±0.2 | 3.3±0.30 |
| | | | 3.1±0.1 | |
| | | | 3.9±0.2 | |
| | | | 2.5±0.1 | |
| | | | 3.0±0.1 | |
| NZ 14A | 4-11 µm polymineral | pIRIR$_{225}$ | 2.7±0.2 | 3.0±0.2 |
| | | | 2.8±0.1 | |
| | | | 3.0±0.2 | |
| | | | 2.6±0.2 | |
| | | | 3.7±0.2 | |
| NZ 13A | 63-90 µm K-feldspar | pIRIR$_{225}$ | 2.7±0.1 | 2.9±0.1 |
| | | | 2.9±0.1 | |
| | | | 3.0±0.1 | |
| NZ 14A | 63-90 µm K-feldspar | pIRIR$_{225}$ | 1.1±0.05 | 1.3±0.1 |
| | | | 1.5±0.04 | |
| | | | 1.3±0.03 | |
| NZ 13A | 4-11 µm polymineral | pIRIR$_{290}$ | 7.6±0.3 | 10.0±1.0 |
| | | | 11.4±0.4 | |
| | | | 10.6±0.4 | |
| | | | 12.5±0.5 | |
| | | | 7.6±0.2 | |
| NZ 14A | 4-11 µm polymineral | pIRIR$_{290}$ | 10.4±0.4 | 10.8±0.9 |
| | | | 12.3±0.5 | |
| | | | 12.4±0.5 | |
| | | | 7.4±0.3 | |
| | | | 11.4±0.5 | |
| NZ 13A | 63-90 µm K-feldspar | pIRIR$_{290}$ | 9.7±0.4 | 9.0±0.3 |
| | | | 8.9±0.3 | |
| | | | 8.6±0.2 | |
| NZ 14A | 63-90 µm K-feldspar | pIRIR$_{290}$ | 5.2±0.2 | 4.9±0.2 |
| | | | 4.6±0.3 | |
| | | | 5.0±0.2 | |

**Reviewer:** *One fairly big issue is that there is limited reporting of what landform the luminescence samples were collected from. The locations are given on the aerial photo, but I cannot determine if they were collected from sediment in the channel or from the walls of the canyon, etc. Because of this, it is hard to interpret the interpretations of partial bleaching fully. As the pIR290 ages overestimate the pIR225 ages, there is likely to be partial bleaching, but how much? Are the ages themselves suspect or not? How do you know? Do you need a lower temperature pIR signal to get a better bleached age?*

**Answer:**

The samples were collected from the flanks of the two largest canyons, just above the boundary between the gravels below and loess above (see photographs below, which will be included in revised manuscript).

[Figure]

NZ13A

[Figure]

NZ14 A

Indeed, the partial bleaching is an important concern especially as pIRIR signals are known to be more difficult to bleach. Existing studies have shown that the pIRIR residual doses can range between a few Gray (<2 Gy) to 10-20 Gy or even more (Thiel et al., 2011; Stevens et al., 2011;

Buylaert et al., 2011, 2012; Murray et al., 2012, 2014; Yi et al., 2016, 2018; Veres et al., 2018; Avram et al., 2020). Several studies reported a slight dependency between the natural and residual dose, the latter increasing with equivalent dose (Sohbati et al., 2012; Buylaert et al., 2012; Murray et al., 2014). Long term laboratory bleaching experiments have been conducted on loess samples from northern (Yi et al., 2016) and south-eastern (Yi et al., 2018) China. They investigated the degree to which the $pIRIR_{290}$ signal was bleachable by exposing several samples in a Honle SOL2 simulator over various periods. For the loess samples from northern China, they have shown that a constant (or highly difficult to bleach) residual $pIRIR_{290}$ signal (corresponding to a dose of 6.2±0.7 Gy) is reached after 300 h, while for the loess samples from south-eastern China a constant residual dose of 4±1 Gy was obtained after 300 h bleaching in solar simulator.

A similar experiment was conducted by our team for a sample collected from a nearby site from South Island, New Zealand (unpublished data). Groups of five polymineral fine grains aliquots were exposed to sunlight (to window – only during daylight) over different periods of time. The residual doses were measured using both pIRIR protocols. In the case of the $pIRIR_{225}$ protocol, a constant residual dose was obtained after 48 h exposure to sunlight while in the case of $pIRIR_{290}$ protocol, the residual dose appeared consistent with the constant dose after a bleaching period of 96 h. Also, we have found that the residual dose decrease to 15% of equivalent dose for $pIRIR_{225}$ after 20 minutes exposure to sunlight. We state that these results are not reported yet. However, this is in our view evidence that $pIRIR_{225}$ signals are bleached.

Also, we have tried to measure pIRIR signals using lower temperature on a nearby site from New Zealand, South Island (unpublished data). We have applied the MET protocol (Fu and Li, 2013) on polymineral fine grains extracted from two samples with no success. In the next figures we show the natural pIRIR signals measured using different IR stimulation temperatures (80°C, 110°C, 140°C and 170°C).

Sample 1 with an equivalent dose of ~10 Gy.

[Figure]

Stimulation IR temp 80°C                    Stimulation IR temp 110°C

[Figure]

Sample 2 with an equivalent dose of ~100 Gy.

[Figure]

Moreover, the corrected IRSL signal does not grow properly with the magnitude of the irradiation dose; therefore, the construction of the dose response curve cannot be achieved.

Given these poor results on samples collected from a nearby site, the authors decided that such measurements will bring no added value for dating the samples investigated in this study.

*References:*

Alexanderson, H. & Murray, A. S. 2012. Luminescence signals from modern sediments in a glaciated bay, NW Svalbard. Quaternary Geochronology 10, 250–256.

Avram, A., Constantin, D., Veres, D., Kelemen, S., Obreht, I., Hambach, U., Marković, S.B., Timar-Gabor, A., 2020. Testing polymineral post-IR IRSL and quartz SAR-OSL protocols on

Middle to Late Pleistocene loess at Batajnica, Serbia. Boreas, https://doi.org/10.1111/bor.12442. ISSN 0300-9483.

Buylaert, J.P., Thiel, C., Murray, A.S., Vandenberghe, D., Yi, S.W., Lu, H.Y., 2011. IRSL and post IR-IRSL residual doses recorded in modern dust samples from the Chinese loess plateau. Geochronometria 38, 432–440.

Buylaert, J.P., Jain, M., Murray, A.S., Thomsen, K.J., Thiel, C., Sohbati, R., 2012. A robust feldspar luminescence dating method for Middle and Late Pleistocene sediments. Boreas 41, 435–451.

Dietze, M., Kreutzer, S., Fuchs, M.C., Burrow, C., Fischer, M., Schmidt, C., 2013. A practical guide to the R package luminescence. Ancient TL 31, 11-18.

Fu, X. & Li, S. H. 2013: A modified multi-elevated-temperature post-IR IRSL protocol for dating Holocene sediments using K-feldspar. Quaternary Geochronology 17, 44–54.

Huntley, D. J. & Lamothe, M. 2001. Ubiquity of anomalous fading in K-feldspars and the measurement and correction for it in optical dating. Canadian Journal of Earth Sciences 38, 1093–1106.

Kreutzer, S., Schmidt, C., DeWitt, R., Fuchs, M., 2014. The a-value of polymineral fine grain samples measured with the post-IR IRSL protocol. Radiation Measurements 69, 18-29.

Lowick, S. E., Trauerstein, M. & Preusser, F. 2012. Testing the application of post IR-IRSL dating to fine grain waterlain sediments. Quaternary Geochronology 8, 33–40.

Li, S.-H., Wintle, A.G., 1992. A global view of the stability of luminescence signal from loess. Quaternary Science Reviews 11, 133-137

Murray, A. S., Schmidt, E. D., Stevens, T., Buylaert, J.-P., Markovi_c, S. B., Tsukamoto, S. & Frechen, M. 2014. Dating Middle Pleistocene loess from Stari Slankamen (Vojvodina, Serbia) — Limitations imposed by the saturation behaviour of an elevated temperature IRSL signal. Catena 117, 34–42.

Murray, A.S., Thomsen, K.J., Masuda, N., Buylaert, J.P., Jain, M., 2012. Identifying wellbleached quartz using the different bleaching rates of quartz and feldspar luminescence signals. Radiation Measurements 47, 688–695.

Qin, J. T. & Zhou, L. P. 2012. Effects of thermally transferred signals in the post-IR IRSL SAR protocol. Radiation Measurements 47, 710–715.

Roberts, H. M. 2012. Testing Post-IR IRSL protocols for minimising fading in feldspars, using Alaskan loess with independent chronological control. Radiation Measurements 47, 716–724.

Rahimzadeh, N., Thiel, C., Sprafke, T., Frechen, M. Conference abstract. DLED 2019, Bingen, Germany.

Stevens, T., Marković, S.B., Zech, M., Sümegi, P., 2011. Dust deposition and climate in the Carpathian Basin over an independently dated last glacial-interglacial cycle. Quaternary Science Reviews 30, 662–681.

Sohbati, R., Borella, J., Murray, A., Quigley, M., Buylaert, J.-P., 2016. Optical dating of loessic hillslope sediments constrains timing of prehistoric rockfalls, Christchurch, New Zealand. Journal of Quaternary Science 31, 678-690.

Thiel, C., Buylaert, J.-P., Murray, A. S. & Tsukamoto, S. 2011a. On the applicability of post-IR IRSL dating to Japanese loess. Geochronometria 38, 369–378.

Thiel, C., Buylaert, J.P., Murray, A.S., Terhorst, B., Hofer, I., Tsukamoto, S., Frechen, M., 2011b. Luminescence dating of the Stratzing loess profile (Austria) – testing the potential of an elevated temperature post-IR IRSL protocol. Quaternary International 234, 23–31.

Thiel, C., Horv_ath, E. & Frechen, M. 2014. Revisiting the loess/ palaeosol sequence in Paks, Hungary: a post-IR IRSL based chronology for the 'Young Loess Series'. Quaternary International 319, 88–98.

Tsukamoto, S., Jain, M., Murray, A., Thiel, C., Schmidt, E., Wacha, L., Dohrmann, R., Frechen, M., 2012. A comparative study of the luminescence characteristics of polymineral fine grains and coarse-grained K- and Na- rich feldspars. Radiat. Meas. 47, 903-908.

Veres, D., Tecsa, V., Gerasimenko, N., Zeeden, C., Hambach, U. & Timar-Gabor, A. 2018. Short-term soil formation events in last glacial east European loess, evidence from multi-method luminescence dating. Quaternary Science Reviews 200, 34–51.

Yi,S., Buylaert, J.-P.,Murray, A.S., Lu, H., Thiel,C.&Zeng, L. 2016. A detailed post-IR IRSL dating study of the Niuyangzigou loess site in northeastern China. Boreas 45, 644–657.

Yi, S.W., Buylaert, J.P., Murray, A.S., Lu, H.Y., Thiel, C., Zeng, L., 2016. A detailed post-IR IRSL dating study of the Niuyangzigou loess site in northeastern China. Boreas 45, 644–657.

Yi, S., Li, X., Han, Z., Lu, H., Liu, J, Wu, J., 2018. High resolution luminescence chronology for Xiashu Loess deposits of Southeastern China. Journal of Asian Earth Sciences 155, 188-197.

---

## Short Comment (SC3) · 14 Jul 2020

**"The Landscape evolution modeling, on which the authors' interpretations rely on heavily, is wrong. Details are given further on in this review."**

In the Discussion, we relied on the landscape evolution model to:

- Support inferences from other observations
- Suggest that box canyon formation is best represented by a linear diffusive model
- Estimate the ratio of volume of water to the volume of eroded material

The landscape evolution modelling was a useful addition to the study, but the study's objectives could be addressed without relying on landscape evolution modelling.

We would like to point out that re-running the landscape evolution model simulations as described below would take ~30 days.

**"The authors use two models to simulate the erosion of the Canterbury Coast. The first is the stream power model and the second is a linear diffusion model. Neither of these is likely appropriate to simulate the erosion here, nor do the authors provide the necessary analysis to justify their use. First, the stream power model is really for detachment-limited landscapes. However, the authors do not make a case that the erosion and sediment transport conditions here justify the use of the stream power model. Particularly, as the presence of major slope failures and alluvium in the channels suggests that a significant sediment cover effect may be happening here."**

In the paper we were testing which model, between stream power and linear diffusion, better fits the observations to infer the geomorphic process responsible for canyon formation. We agree with the reviewer that the stream power model is not appropriate to represent the evolution of the box canyons. We can make this clearer in the methodology and discussion.

**"Next, the authors use a linear diffusion model as another endmember. However, with the presence of major slope failures, which are significantly non-local and thus not diffusive, it is hard to justify its use and the authors do not provide this justification in the manuscript. The discussion section completely disregards this fact and overall is not convincing."**

The mass wasting flux is taken to be linearly proportional to the topographic slope at low slopes. A non-linear component can be incorporated to take into account episodic landsliding processes that occur on steeper slopes where the traditional linear formulation may break down.

The diffusion model (Howard, 1994, 2007; Forsberg-Taylor et al., 2004; Luo and Howard, 2008; Barnhart et al., 2009, Boatwright and Head, 2019) can be expressed as:

$$\frac{\partial \eta}{\partial t} = D_l S + D_n \left( \frac{1}{1 - (\frac{S}{S_t})^2} - 1 \right)$$

where t is time [T], $D_l$ and $D_n$ are the diffusion coefficients [L 1-3m T m-1], and $S_t$ is the threshold value or critical slope gradient. S is:

$$S = \nabla^2 \eta$$

Where S is slope (dimensionless) and η is the topographic elevation [L].

If $S << S_t$, then the non-linear part of the equation can be considered negligible (i.e. a very small value compared to the linear part), converting the equation above into a linear diffusion model (equation 9).

$$\frac{\partial \eta}{\partial t} = D_l S$$

The slope of the Canterbury Plains is close to 0°, and we assume that the threshold value or critical slope gradient is much higher than the 70° slope of the cliff.

**"Another problem in the methodology occurs with the values used in the model. One problem is when the authors equate the values for K and D in the stream power model and the linear diffusion model. These values represent very physically processes and are likely very different, by orders of magnitude in some cases. Their use of also assuming that K/D is proportional to the seepage flux and surface water shear stress, while I am open to the idea, isn't backed up with a reference or proof of concept."**

We used similar values for K and D for the sake of simplicity and to easily compare the results. We will use different values (based on sensitivity analyses – see below) in the revised simulations.
The assumption that K/D is proportional to the seepage flux and surface water shear stress is based on Howard (1988) and Luo and Howard (2008). This will be clarified in the text.

**"Next the authors state that they obtain the values for M and Tau_t "by trial and error." What does this mean? Did they just pick values until they obtained behavior they wanted?**
**Because the authors picked values by trial and error, rather than by constraining them from some means, it is not clear if the modeling results presented are meaningful.**
**Normally, when one does not know what input parameters should be, a sensitivity analysis is needed to evaluate the outcomes from a full range of possible values for a specific parameter. Finally, there is no sensitivity analysis given on how model outputs change with change in the input parameters. This is needed to interpret the results from landscape evolution modeling."**

We will carry out a sensitivity analyses that we will include in the revised version of the manuscript.

**"On line 297, it says that the authors are testing the models, but these methods are not really a formal test of these LEMs as currently written."**

This should have been written to indicate that we were using the models to test which erosional process forms the canyons, rather than testing the models themselves. This will be corrected in the revised version of the manuscript.

**"One line 337: Actually, there probably is a way in landlab to incorporate sand lenses, but this would essentially involve developing a new landlab component to do so."**

This is correct. We should have written:
"Since Landlab is a 2-D modeling environment, it was not possible to include 3-D sand lenses in its current version. For this reason, we simplified the geology by considering two types of sedimentary units: sandy gravels with a NW-SE strip of sand."

**"Why is Equation 10 the same as Equation 7?"**

This was a mistake. Equation 10 will be removed.

---

## Short Comment (SC4) · 15 Jul 2020

Dear Reviewer 2,

Thanks for your insightful comments. I have gone through these with my colleagues and we think we can address these in a revised version of the manuscript.

Best wishes,

Aaron

[Figure]

**ESurfD**

Interactive
comment

---

## Author Response (AR1)

**RESPONSE TO REVIEWERS LETTER**

5   Dear Editor:

Attached please find the revised version of the manuscript 'Box canyon erosion along the Canterbury coast (New Zealand): A rapid and episodic process controlled by rainfall intensity and substrate variability' (esurf-2020-29)**.** My co-authors and I would like to thank you and
10   the two reviewers for the thorough review and constructive comments. We think that by taking your suggestions into consideration, the paper has been greatly improved.

What follows is a point-to-point discussion of the reviewers' comments and how these have been addressed.

**Comments made by Reviewer 1:**

1.   *The Landscape evolution modeling, on which the authors' interpretations rely on heavily, is wrong. Details are given further on in this review. Because the evidence provided from*
20   *the landscape evolution modeling is not robust, I fear the authors made need to redo their entire LEM analysis including re-performing model runs, including calibrating their parameter values and conducting sensitivity analyses. I fear that the authors will need more time than is offered during Esurf's review process. As of now, I find the authors statements in the discussion unconvincing.*

25   *The authors use two models to simulate the erosion of the Canterbury Coast. The first is the stream power model and the second is a linear diffusion model. Neither of these is likely appropriate to simulate the erosion here, nor do the authors provide the necessary analysis to justify their use. First, the stream power model is really for detachment-limited landscapes. However, the authors do not make a case that the erosion and sediment*
30   *transport conditions here justify the use of the stream power model. Particularly, as the presence of major slope failures and alluvium in the channels suggests that a significant sediment cover effect may be happening here.*

*Next, the authors use a linear diffusion model as another endmember. However, with the presence of major slope failures, which are significantly non-local and thus not diffusive,*
35   *it is hard to justify its use and the authors do not provide this justification in the manuscript. The discussion section completely disregards this fact and overall is not convincing.*

*Another problem in the methodology occurs with the values used in the model. One problem is when the authors equate the values for K and D in the stream power model and*
40   *the linear diffusion model. These values represent very physically processes and are likely very different, by orders of magnitude in some cases. Their use of also assuming that K/D is proportional to the seepage flux and surface water shear stress, while I am open to the idea, isn't backed up with a reference or proof of concept. Next the authors state that they obtain the values for M and Tau_t "by trial and error." What does this mean? Did they*
45   *just pick values until they obtained behavior they wanted? Because the authors picked values by trial and error, rather than by constraining them from some means, it is not clear if the modeling results presented are meaningful. Normally, when one does not know what*

*input parameters should be, a sensitivity analysis is needed to evaluate the outcomes from a full range of possible values for a specific parameter.*

50 *Finally, there is no sensitivity analysis given on how model outputs change with change in the input parameters. This is needed to interpret the results from landscape evolution modeling. On line 297, it says that the authors are testing the models, but these methods are not really a formal test of these LEMs as currently written.*

*One line 337: Actually, there probably is a way in landlab to incorporate sand lenses, but*
55 *this would essentially involve developing a new landlab component to do so.*

*Why is Equation 10 is the same as Equation 7?*

In the Discussion, we relied on the landscape evolution model to:

60 • Support inferences from other observations
• Suggest that box canyon formation is best represented by a linear diffusive model
• Estimate the ratio of volume of water to the volume of eroded material

The landscape evolution modelling was an interesting addition to the study, but the study's
65 objectives could be addressed without relying on it. In view of the above comments, I have decided to remove the sections related to landscape evolution modelling from the paper. The ratio of volume of water to the volume of eroded material is now estimated using the slope stability model. The inference on the linear diffusive model has been removed.

70 2. *The data presented are not fully synthesized into a clear and complete story or analysis. The challenge here is that it is hard to understand how each of the data the authors present ties into the overall picture of 'box canyon' erosion. As such, it is not clear as the paper is currently written what the contribution of the paper is towards understanding Earth surface dynamics. After reading the paper, the only things I got out of it are that the authors*
75 *did a lot of work collecting a lot of data, but I can't say I understand what the contribution is. Two good examples are the luminescence ages and the geophysical resistivity surveying. The ages are used mostly in a passing way and the geophysical resistivity doesn't seem like it was very successful, so it not clear why it is included?*

80 In our contribution we address three points: (i) how coastal gullies are formed by groundwater erosion, (ii) the role that lithology and permeability play in gully initiation and evolution, and (iii) the temporal scale of gully formation. Because of the reliance on experiments and numerical models, and the paucity of process-based observations and instrumental analyses, previous studies had not thoroughly addressed these issues in the past.
85 By integrating field data with modelling, we show that gully formation is an episodic process associated to groundwater flow that occurs when a threshold rainfall intensity is exceeded. The gullies in our study area are actively eroding, and erosion rates can be as high as 30 m per day. Hydraulically-conductive zones (e.g. relict braided rivers channels, tunnels) and sand lenses exposed at the cliff face control the location of the gullies. We believe that these
90 outcomes are important because: (i) there are only a few places where the mechanisms by which seepage erosion occurs have been clearly defined, and (ii) the results can be used to test and quantify models of gully formation and improve the reconstruction/prediction of landscape evolution by groundwater-related processes.

In the revised version of the manuscript we have updated the Discussion by:

- Subdividing the section into 3 sub-sections that directly address each individual objective
- Referring to the observations, figures and tables in the Results section to support the inferences made in each section

The OSL ages are used to demonstrate that the largest gullies have been inactive in the last 2 ka. This contrasts with the inference by Schumm and Phillips (1986) that they were formed by spillage of water from swamps behind the cliffs in the 19th century. It also suggests that they may have formed by processes other than groundwater flow, and may explain why the longer gullies have different morphometrics in comparison to the smaller (younger) gullies.

The comment about the geophysical resistivity is addressed in point 9.

3. *The term Box Canyon doesn't bring to mind the features that the study focuses on. I understand box canyons to be canyons with vertical walls, flat bottoms, and generally ingress and egress is only possible through the mouth of the canyon. The classic box canyon are the narrow canyons typically associated with the arid western United States. I think what the authors are studying are actually gullies that progress into canyons when they collect enough drainage area. As the authors note in Lines 383-385, the younger and lower-discharge/drainage areas/stream orders of these features have V and U shaped cross sectional profiles. Whereas the typical box canyons I have seen in the field continue to have flat bottoms and vertical walls even at low stream orders. I think this will be a problem for the authors in that researchers looking for information on box canyons will not consider this work relevant, and those studying coastal gully erosion will miss this paper due to the title and mislabeling of the features of interest. Instead, I think the authors should rewrite this paper while avoiding the use of the term box canyon. Either referring to these as coastal cliff gullies, headward cliff erosion, or some new term the author's come up with.*

The reviewer's point is valid. We now refer to these landforms as coastal gullies in the revised version of the manuscript. We have also revised the Introduction to include a section on the state of the art and gaps in knowledge related to coastal gully research.

4. *These values don't seem all that great to me. I generally try changing my protocol if the dose recovery ratio comes out worse than 1.10 or 0.9. What could be causes of this not-great dose recovery? Is this typical for feldspar from New Zealand?*

So far, several studies reported poor dose recovery results for pIRIR$_{290}$ (e.g. Stevens et al., 2011; Thiel et al., 2011, 2014; Lowick et al., 2012; Roberts et al., 2012; Murray et al., 2014, Veres et al., 2019; Avram et al., 2020). Usually, these poor dose recovery ratios were greater than unity. On the other hand, Sohbati et al. (2016) reported good dose recovery ratios for K-feldspars using pIRIR$_{290}$ extracted from palaeorockfall boulders from South Island, New Zealand.

Qin and Zhou (2012) suggested that the dose recovery ratio is dependent on the test dose size. Yi et al. (2016) concluded that the best dose recovery ratios are found when the test dose ranges between 15 and 80% of the total dose. Here, following this suggestion, dose recovery measurements were carried out using a test dose ranging between 20 and 60% of the total

dose. On the other hand, a recent study by Avram et al. (2020) reported poor dose recovery ratio using a test dose of 41% of the total dose on polymineral fine grains extracted from Serbian loess.

As the reviewer suggested, two protocols are actually being used, namely pIRIR$_{290}$ and pIRIR$_{225}$. As presented in the manuscript "the ratios between the recovered and given dose for polymineral fine grains are 1.11±0.03 (NZ13A) and 1.01±0.03 (NZ14A) using piRIR$_{225}$ protocol and 1.14±0.05 (NZ13A) and 1.03±0.06 (NZ14A) using pIRIR$_{290}$ protocol. In the case of coarse K-feldspars, the dose recovery ratios are 0.96±0.01 (NZ13A) and 0.87±0.03 (NZ14A) using pIRIR$_{225}$ protocol, while those obtained by applying pIRIR$_{290}$ protocol are 1.17±0.06 (NZ13A) and 1.16±0.04 (NZ14A)." Please note that in the case of the pIRIR$_{225}$ protocol, the results are generally satisfactory – they are within 1σ uncertainty and they are consistent with the 0.9-1.1 interval. We have stated that "We conclude that the overall behaviour of the pIRIR$_{225}$ protocol is satisfactory whereas the pIRIR$_{290}$ protocol overestimates the given dose by ~ 17%."

Some authors (e.g Stevens et al., 2011; Alexanderson and Murray, 2012) concluded that poor dose recovery ratios can be the result of the incorrect residual dose estimation.

In order to avoid potential complications related to laboratory bleaching of the natural samples, the dose recovery test could be performed by adding a beta dose on top of the natural dose. Then, the dose recovery ratios are obtained by dividing the measured dose by the sum of the natural and the given dose.

Such a dose recovery test was performed in the case of pIRIR$_{290}$ protocol by adding a beta dose of 100 Gy on top of natural dose. The results are presented in column 4 of the table below and compared with the previously reported values stated above and presented in the manuscript (column 3).

| Sample code | Grain size | Recovered/given ratio (residual dose subtraction) | Recovered/given ratio (100 Gy on top of natural) |
|---|---|---|---|
| NZ13A | 4-11 μm polymineral | 1.14±0.05 | 1.12±0.05 |
| NZ14A | 4-11 μm polymineral | 1.03±0.06 | 1.15±0.03 |
| NZ13A | 63-90 μm K-feldspar | 1.17±0.06 | 1.03±0.03 |
| NZ14A | 63-90 μm K-feldspar | 1.16±0.04 | 1.02±0.02 |

For coarse K-feldspars, the dose recovery ratio improved from 1.17±0.06 to 1.03±0.03 for sample NZ13A and from 1.16±0.04 to 1.02±0.02 for sample NZ14A. In the case of polymineral fine grains, the dose recovery ratio for sample NZ13A was still slightly overestimated (1.12±0.05), while for sample NZ14A, the dose recovery ratio increased from 1.03±0.06 to 1.15±0.03. As such there is no clear trend. We can only conclude that while the results are not ideal, they are acceptable, and the used protocols are providing the best attainable results for these samples given state-of-the-art available methods (see also answers below regarding changing the preheat temperature).

    5.    *Those g-values might not be insignificant. If you perform the fading correction on the k-feldspar for NZ14A, do the ages agree with the polymineral?*

The g-values are not significant in our view. We have discussed this in detail in Avram et al.
(2020), where we are presenting more extended fading measurements, including on
185  calibration quartz. Consequently, as we do not consider these short-term measurements to be
reliable and we do not consider the values to be significant, we are not presenting corrected
ages. For the sake of the exercise, following the reviewer's suggestion, we have calculated the
corrected K-feldspars ages (pIRIR$_{225}$) for both samples NZ13A and NZ14A. The results are
presented in the next table:

190

| Sample code | g-value (%/decade) | Uncorrected Age (ka) | | | | | Corrected Age (ka) |
|---|---|---|---|---|---|---|---|
| | K-feldspar (pIRIR$_{225}$) | pfg (pIRIR$_{225}$) | K-feldspar (pIRIR$_{225}$) | pfg (pIRIR$_{290}$) | K-feldspar (pIRIR$_{290}$) | | K-feldspar (pIRIR$_{225}$) |
| NZ 13A | 0.60±0.51 | 16±0.1 | 20.1±1.5 | 20.9±2.0 | 26.2±2.1 | | 20.8±1.5 |
| NZ 14A | 0.85±0.09 | 4.6±0.4 | 1.9±0.1 | 6.0±0.7 | 3.1±0.3 | | 2.0±0.1 |

The corrected pIRIR ages were calculated using the R Luminescence-package (Dietze et al.,
2013) according to Huntley and Lamothe (2001) method.

195  As can be seen, there is no significant difference in the ages obtained.

      6.    *Perhaps the discrepancy in age between the polymineral fine grain and the k-feldspar (for NZ14A) is due to the presence of unknown minerals in the polymineral aliquots? It is possible that there are minerals in there with unknown pIRIR properties that might be*
200          *transferring charge in unexpected ways?*

This is indeed a reasonable assumption that is worth further study. The polymineral fraction
consists of a mixture of different kinds of minerals but it is assumed that the measured
emission in the violet-blue band is predominantly caused by feldspar minerals (Tsukamoto et
205  al., 2012; Kreutzer et al., 2014). Li and Wintle (1992) investigated the stability of luminescence
signal of polymineral fine grains in comparison with those from K- and Na- rich feldspars, all
extracted from loess. They found that the thermal stability of the IRSL signals from
polymineral fine grains extracted from different areas around the world is similar, and the
IRSL signal from fine grains is less stable than the signal of K-feldspars. Tsukamoto et al.
210  (2012) found that IRSL and pIRIR signals from polymineral fine grains are less stable than K-
feldspar sample when a lower preheat temperature (260–300 °C) is used, but the stability
among different types of feldspars becomes similar when a higher preheat temperature (>320
°C) is included. From a dating point of view, they suggested that the IRSL in polymineral fine
grains behaves similarly to the Na-feldspars signal, rather than to the K-feldspar's signal. To
215  our knowledge, there are few, if any, published luminescence dating studies that documented
both polymineral and K-feldspars ages obtained by using pIRIR methods. Rahimzadeh et al.,
(2019 - conference abstract) reported polymineral fine grains and K-feldspars ages using
pIRIR$_{225}$ protocol on Bavarian loess. They have found that the fading corrected ages for the

two different grain sizes agree within errors for most of the samples, observing a larger discrepancy for samples with an age above 100 ka. These kinds of studies are worth pursuing, however; in our view it would be better to attempt such studies on sites where independent age control or at least well-behaved quartz chronologies are available.

Here, we have further investigated the growth curves of polymineral fine grains and K-feldspars (see figure below). As can be seen, the growth curves overlap over the dose range investigated.

[Figure]

7.     *One other thing that might be useful is if the authors could report the distribution of the residual dose. If there is large variance, maybe this could explain the discrepancy of the polymineral and feldspar ages.*

Residual doses measured for each individual aliquot are reported in the table below. There is no significant scatter in between the measured residual doses for each aliquot, and certainly not a difference that could account for the reported age discrepancy. Therefore, we conclude that this scenario cannot explain the observed discrepancy between polymineral and K-feldspars ages.

| Sample code | Grain size | protocol | Residual dose | Average residual dose |
|---|---|---|---|---|
| NZ 13A | 4-11 μm polymineral | pIRIR$_{225}$ | 4.2±0.2 | 3.3±0.30 |
| | | | 3.1±0.1 | |
| | | | 3.9±0.2 | |
| | | | 2.5±0.1 | |
| | | | 3.0±0.1 | |
| NZ 14A | 4-11 μm polymineral | pIRIR$_{225}$ | 2.7±0.2 | 3.0±0.2 |
| | | | 2.8±0.1 | |
| | | | 3.0±0.2 | |
| | | | 2.6±0.2 | |
| | | | 3.7±0.2 | |
| NZ 13A | 63-90 μm K-feldspar | pIRIR$_{225}$ | 2.7±0.1 | 2.9±0.1 |
| | | | 2.9±0.1 | |
| | | | 3.0±0.1 | |
| NZ 14A | 63-90 μm K-feldspar | pIRIR$_{225}$ | 1.1±0.05 | 1.3±0.1 |
| | | | 1.5±0.04 | |
| | | | 1.3±0.03 | |
| NZ 13A | 4-11 μm polymineral | pIRIR$_{290}$ | 7.6±0.3 | 10.0±1.0 |
| | | | 11.4±0.4 | |
| | | | 10.6±0.4 | |
| | | | 12.5±0.5 | |
| | | | 7.6±0.2 | |
| NZ 14A | 4-11 μm polymineral | pIRIR$_{290}$ | 10.4±0.4 | 10.8±0.9 |
| | | | 12.3±0.5 | |
| | | | 12.4±0.5 | |
| | | | 7.4±0.3 | |
| | | | 11.4±0.5 | |
| NZ 13A | 63-90 μm K-feldspar | pIRIR$_{290}$ | 9.7±0.4 | 9.0±0.3 |
| | | | 8.9±0.3 | |
| | | | 8.6±0.2 | |
| NZ 14A | 63-90 μm K-feldspar | pIRIR$_{290}$ | 5.2±0.2 | 4.9±0.2 |
| | | | 4.6±0.3 | |
| | | | 5.0±0.2 | |

240

8. *One fairly big issue is that there is limited reporting of what landform the luminescence samples were collected from. The locations are given on the aerial photo, but I cannot determine if they were collected from sediment in the channel or from the walls of the canyon, etc. Because of this, it is hard to interpret the interpretations of partial bleaching fully. As the pIR290 ages overestimate the pIR225 ages, there is likely to be partial bleaching, but how much? Are the ages themselves suspect or not? How do you know? Do you need a lower temperature pIR signal to get a better bleached age?*

The samples were collected from the flanks of the two largest gullies, just above the boundary between the gravels below and loess above (see revised Fig. 1).

Indeed, partial bleaching is an important concern especially as pIRIR signals are known to be more difficult to bleach. Existing studies have shown that the pIRIR residual doses can range between a few Gray (<2 Gy) to 10-20 Gy or even more (Thiel et al., 2011; Stevens et al., 2011; Buylaert et al., 2011, 2012; Murray et al., 2012, 2014; Yi et al., 2016, 2018; Veres et al., 2018; Avram et al., 2020). Several studies reported a slight dependency between the natural and residual dose, the latter increasing with equivalent dose (Sohbati et al., 2012; Buylaert et al., 2012; Murray et al., 2014). Long term laboratory bleaching experiments have been conducted on loess samples from northern (Yi et al., 2016) and south-eastern (Yi et al., 2018) China. They investigated the degree to which the $pIRIR_{290}$ signal was bleachable by exposing several samples in a Honle SOL2 simulator over various periods. For the loess samples from northern China, they have shown that a constant (or highly difficult to bleach) residual $pIRIR_{290}$ signal (corresponding to a dose of 6.2±0.7 Gy) is reached after 300 h, while for the loess samples from south-eastern China a constant residual dose of 4±1 Gy was obtained after 300 h bleaching in a solar simulator.

A similar experiment was conducted by our team for a sample collected from a nearby site from South Island, New Zealand (unpublished data). Groups of five polymineral fine grains aliquots were exposed to sunlight (to window – only during daylight) over different periods of time. The residual doses were measured using both pIRIR protocols. In the case of the $pIRIR_{225}$ protocol, a constant residual dose was obtained after 48 h exposure to sunlight while in the case of $pIRIR_{290}$ protocol, the residual dose appeared consistent with the constant dose after a bleaching period of 96 h. Also, we have found that the residual dose decrease to 15% of equivalent dose for $pIRIR_{225}$ after 20 minutes exposure to sunlight. We state that these results are not reported yet. However, this is in our view evidence that $pIRIR_{225}$ signals are bleached.

Also, we have tried to measure pIRIR signals using lower temperature on a nearby site from New Zealand, South Island (unpublished data). We have applied the MET protocol (Fu and Li, 2013) on polymineral fine grains extracted from two samples with no success. In the next figures we show the natural pIRIR signals measured using different IR stimulation temperatures (80°C, 110°C, 140°C and 170°C).

Sample 1 with an equivalent dose of ~10 Gy.

[Figure]

Sample 2 with an equivalent dose of ~100 Gy.

[Figure]

Moreover, the corrected IRSL signal does not grow properly with the magnitude of the irradiation dose; therefore, the construction of the dose response curve cannot be achieved.

285

290

Given these poor results on samples collected from a nearby site, the authors decided that such measurements will bring no added value for dating the samples investigated in this study.

9. *The geophysical resistivity doesn't seem like it was very successful, so it not clear why it is included?"*

The inability to fit a 1-D resistivity-vs-depth model to the observed slingram responses, coupled with the observation that the first-time-gate response profiles show considerable along-transect variability (original Figs. 9a-c), suggest that the geo-electrical subsurface structure is strongly heterogeneous within the footprint of the G-TEM transmitter. The geoelectrical heterogeneity suggested by the G-TEM responses could be caused by the resistivity signature of discrete subsurface groundwater conduits, perhaps in the form of cylindrical tubes of irregular cross-section, carrying groundwater seaward across the Canterbury Plains and discharging either at the cliffs or the beach.

This is the key information provided by G-TEM in support of the hypothesis of discrete bodies of groundwater flowing along preferential seepage pathways and supporting the growth and evolution of gullies along the coastal cliffs.

Work is currently underway, as part of a different study, on 2-D forward modelling of the slingram responses; our preliminary results (see new section in Supplementary Materials) indicate that the G-TEM data can be explained by the aforementioned electrically conductive groundwater conduits. It suffices to say here that, to depths of several tens of metres, the subsurface landward of the gullies is certainly not one-dimensional from the perspective of electrical resistivity.

Please see also response to comments 12-24.

**Comments made by Reviewer 2:**

10. *The paper is interesting but the geophysical approach needs more attention and some more information. Therefore, even if the authors did a nice job, they should rewrite the geophysical part or eliminate it, in order to work more on the data elaboration.*

11. *Line 153: Why the authors made the survey in these dates? I suggest to introduce if there were some climatic phenomena or some motivation about these specific dates.*

The UAV surveys were carried out after rainfall events. This is now clarified in section 3.1.2.

12. *Line 165: the investigation depth depends to the electrical characteristic of the subsoil. How you estimated this depth of investigation?*

The sentence "*All soundings were collected in the 20-gate mode with an acquisition interval of $6 \times 10^{-6}$ s to $8 \times 10^{-4}$ s (after ramp-off), corresponding to investigation depths of ~80 m*" was removed and replaced with: "The maximum depth of investigation of the G-TEM system is given approximately by the formula $d=8.94L^{0.4}\rho^{0.25}$ where $L$ (m) is the TX loop size and $\rho$ ($\Omega$m) is the

upper layer resistivity (Geonics 2016). Setting ρ=100 Ωm yields a depth of investigation of *d*=71 m, whereas ρ=1000 Ωm yields *d*=126 m. Our investigation depth in New Zealand may be slightly greater than these values since the Geonics formula assumes a 1-turn TX loop carrying current 3 A whereas we used a more powerful combination of 4 turns at 1.5 A. "

13.    *Line 167: I suggest to add some references about*

The sentence "...*using the IXG-TEM software from Interpex Limited"* has been replaced with "using IXG-TEM commercial software (Interpex 2012). This is a standard 1-D TDEM inversion code that has been successfully used for coastal hydrogeophysical studies (Pondthai et al., 2020)."

14.    *Figure 8b: Which is the graph of a model with lower resistive layers? In example, did you tried to use the same of the 1D inversion result?*

After the sentence on line 464, *"As a result, we cannot trust 1-D inversions of the slingram-mode data in such a 3-D geological environment"*, the following sentence was added: "We did not try to use the 1-D inversion software to further analyse and interpret the G-TEM data. However, even though the individual slingram-mode responses cannot be fit reliably by a 1-D model, we can still analyse lateral changes in the observed response curves along the slingram profiles to reveal information about subsurface heterogeneity; this is elaborated below."

15.    *Figure 8b: How have you produced these models? I think it is important to describe them or to add some references about.*

After the sentence on line 476: *"Figure 8b displays a 1-D model (A) that contains a conductive layer of 100 Ωm between 10-20 m depths in a homogeneous 1000 Ωm background"*, the following sentences were included: *"*The 1-D model A in Fig. 8b is motivated by the inversion results of deep-penetrating 40 × 40 m TX loop TDEM soundings carried out on top of the cliffs several tens of meters inland (Weymer et al, submitted manuscript), which revealed such a conductive zone at these depths. Unlike the slingram profiles, the deeper-penetrating, larger-loop sounding curves are readily fit by a 1-D model.*"*

16.    *Line 473: Which are the advantages to use this approach? I suggest to describe it in detail or add some references about.*

After the sentence on line 473: *"Instead of performing 1-D inversions, we present time-gate plots for all three transects"*, the following sentences were added: *"*A time-gate plot is defined as a graph of the observed G-TEM voltage response, evaluated at a particular time-gate, as a function of position along a profile. Time-gate plots are a useful alternative to explore the lateral variability of the G-TEM response along a profile in the event that the sounding curves at individual stations cannot be fit with 1-D models. It is presumed that variability in a time-gate plot is correlated with lateral heterogeneity in the subsurface geoelectrical structure, since a 1-D Earth structure would yield no spatial variability in a time-gate plot.*"*

17.    *Line 477: Larger amplitude?  What do you mean with larger?*

The sentence on line 477: "*This model generates a G-TEM slingram response that has a substantially larger amplitude at all time gates than does the model (B) without the conductive layer*" was changed to "This model generates a G-TEM slingram response that has a substantially larger ramp-off voltage amplitude at all time gates than does the model (B) without the conductive layer, as shown in Fig. 7b."

    18.    *Line 478: Before to write this sentence I suggest to add a paragraph on your approach, in order to explain this results. I suggest to add some references about this kind of results.*

After the sentence on line 478: "*Thus we regard an enhancement of response at the first time gate as indicative of a conductive zone at depth beneath the slingram station*", we added the following sentences: "The spatial analysis of time-gate plots is not a conventional approach in time-domain electromagnetics but it is somewhat analogous to the spatial analysis of apparent resistivity profiles in frequency-domain electromagnetics using terrain conductivity meters (e.g. Weymer et al. 2016), based on the idea that the G-TEM response at a fixed time-gate carries information similar to that of a terrain conductivity meter response at a fixed frequency."

    19.    *Line 481: The description is not clear. The used plots are associated to "first-time-gate", what do you mean? In the model you use a different plot than for the acquired data, I suggest to use the same approach in order to understand the used methodologies, which is not common.*

After the sentence on line 478: "*The first-time-gate profile of transect May15-1 is located upslope of small but recently eroded box canyons (Fig. 9a)*" we added the following sentence: "In this figure, the 'first-time-gate profile' is a plot as a function of distance along the transect of the G-TEM ramp-off voltage at time gate number 1 - as shown, for example, in Fig. 7b - at the first sampled point of the transient response immediately after the TX current has been switched off."

    20.    *Line 489: With all this "not shown" results, it is hard to understand your approach and the final results.*

The sentence on line 488 from: "*The fluctuating signals remain similar in shape for time-gates 2 to 6 (not shown)*" was changed to "The time-gate plots for gates 2 to 7 remain similar in shape to that of the time-gate-1 plot and hence are not shown. After time-gate 7, the time-gate plots start to lose coherence due to the low signal-to-noise ratio of the decaying RX voltage at late times after TX ramp-off."

    21.    *Line 493: From my point of view, this observation is not clear taking in account the GTEM results. I suggest to improve the elaboration of them. May be these results are not so consistent and compatible with the geological ones.*

The G-TEM results are now more thoroughly explained, as shown in our response to the previous comments. We also include the following sentences in the Discussion (section 5.2): "With regards to the G-TEM slingram time-gate plots (Fig. 8), we interpret the higher-amplitude responses on the time-gate-1 plots that are preferentially located upslope of recently active gullies as zones of relatively high electrical conductivity in the subsurface at

depths of ~10 m. These zones are suggestive of buried groundwater conduits made up of gravel and/or sandy units (e.g. Weymer et al., in review), or tunnels formed by sub-surface groundwater flow in sand units. Further analysis of the G-TEM data, including 2-D modelling and inversion, is required to ascertain the sub-surface hydraulic geometry responsible for the along-profile amplitude variations. This is elaborated further in the Supplementary Materials... If seaward-directed groundwater conduits are responsible for the location of gullies, the G-TEM results predict that, along the Canterbury coast, we should generally observe active gully development downslope of peaks in slingram time-gate plots. If this is the case, G-TEM could be used to identify locations of incipient and even future gully development."

22. *Figure 9: I suggest to define the x legend...If I think it is meters.*

The sentence "A yellow line marks a slingram transect, the length of which can be determined from the scale bar" was added to the caption of the original Fig. 9.

23. *Line 654: From my point of view, this geophysical results are not so clear. What do you mean with "higher amplitude"? I think the paper needs more detail on the geophysical approach, in order to write this sentence.*

The sentence on line 654: "*We interpret the higher amplitude response in G-TEM profiles located upslope of recently active box canyons as buried groundwater conduits...*" was changed to " With regards to the G-TEM slingram time-gate plots (Fig. 8), we interpret the higher-amplitude responses on the time-gate-1 plots that are preferentially located upslope of recently active gullies as zones of relatively high electrical conductivity in the subsurface at depths of ~10 m. These zones are suggestive of buried groundwater conduits..."

24. *Line 656: I agree with you. Therefore, I suggest to obtain these results and after to add the geophysical data in this paper.*

After the sentence on line 656: "*Further analysis of the G-TEM data, including 2-D modelling and inversion, is required to ascertain the sub-surface hydraulic geometry responsible for the along-profile amplitude variations*", we added the sentence "This is elaborated further in the Supplementary Materials." A new related section has been included in the Supplementary Materials that describes the G-TEM slingram modelling in detail.

We hope that we have managed to address all your concerns and that you are satisfied with the above changes. I also kindly ask the editor to update the title of the manuscript to:

**Groundwater erosion of coastal gullies along the Canterbury coast (New Zealand): A rapid and episodic process controlled by rainfall intensity and substrate variability**

and to remove Roger Clavera-Gispert from the author list and include Phillipe Wernette.

With best regards also on behalf of my co-authors,

Aaron Micallef

[revised manuscript text omitted]

Font: (Default) Times New Roman, 10 pt

| Page 42: [8] Formatted | Microsoft account | 29/07/2020 14:13:00 |
|---|---|---|

Font: (Default) Times New Roman, 10 pt, Font color: Black

| Page 42: [9] Formatted | Microsoft account | 29/07/2020 14:13:00 |
|---|---|---|

Font: (Default) Times New Roman, 10 pt, Font color: Black

| Page 42: [10] Formatted | Microsoft account | 07/08/2020 16:25:00 |
|---|---|---|

Highlight

| Page 42: [11] Deleted | Microsoft account | 30/07/2020 17:01:00 |
|---|---|---|

The shear stress of the water seeping out of the canyon head and flowing along the base of the canyon is high enough to entrain the failed sandy material (shear stress of $7.5 \times 10^{-5}$ in Fig. 15a; Petit et al. (2015))(Julien, 1998), which explains why the floors of box canyons are primarily covered by gravel.

| Page 42: [12] Formatted | Microsoft account | 07/08/2020 16:25:00 |
|---|---|---|

Highlight

| Page 42: [13] Formatted | Microsoft account | 07/08/2020 16:25:00 |
|---|---|---|

Highlight

| Page 42: [14] Formatted | Microsoft account | 07/08/2020 16:25:00 |

Highlight

| Page 42: [15] Formatted | Microsoft account | 07/08/2020 16:25:00 |

Highlight

| Page 42: [16] Formatted | Microsoft account | 07/08/2020 16:24:00 |

Not Highlight

| Page 42: [17] Deleted | Microsoft account | 30/07/2020 15:55:00 |

~~The field observations of box canyon morphology and development are most similar to the LEM results for the high intensity rainfall scenario of the linear diffusive erosion model (Fig. 16). There are two reasons for this: first, as observed in Culling (1963) and Barnhardt et al. (2019), the retrogressive slope failure is best simulated by a diffusive linear model; secondly, our linear diffusive erosion model links the diffusion co-efficient with the shear stress generated by the groundwater. The evolution of the different models and scenarios appears to be influenced by the initial topography – with the cliff preventing the upslope development of erosive features in the stream power law mode, regardless of the recharge values – and the water table height, which depends on permeability and recharge (Figs. 13-16). In the stream power law mode, erosion is linked to surface water, which depends exclusively on groundwater seepage. Since the water table can only intersect the beach or the cliff surface, surface erosion of the plain above the cliff top is not possible (Figs. 13-14). When the stream power law model does not depend on groundwater seepage, the surface erosion does affect the cliff face, but it results in a series of closely spaced and narrow gullies (Fig. 17). The conceptual model, the program configuration and the initial and boundary conditions make the LEM outcomes only applicable to box canyon formation at this site.~~

| Page 42: [18] Formatted | Microsoft account | 30/07/2020 14:41:00 |
Font: (Default) Times New Roman, 10 pt, Font color: Black

| Page 42: [19] Formatted | Microsoft account | 16/07/2020 12:40:00 |

Strikethrough

---

## Author Response (AR2)

**RESPONSE TO REVIEWERS LETTER**

Dear Editor:

Attached please find the revised version of the manuscript 'Groundwater erosion of coastal gullies along the Canterbury coast (New Zealand): A rapid and episodic process controlled
10  by rainfall intensity and substrate variability' (esurf-2020-29)**.** My co-authors and I would like to thank you and the reviewer for your comments.

What follows is a point-to-point discussion of these comments and how these have been addressed.

**Comments made by Editor:**

1.  *Line 70: Now that the landscape evolution modeling has been removed,…*

20  I am sorry but I could not understand this comment.

2.  *Table 1: I do not think this table adds much to the presentation of your study.  You may consider simplifying this to a few sentences in the text.*

25  This modification has been carried out.

3.  *Figure 1: Colorbar on A is very small and difficult to see in b, the dashed boxes are difficult to discern, consider making a different color*

30  These modifications have been carried out.

4.  *Line 165: "Substrate samples were collected for geochronological analysis."*

We have not carried out this modification because the samples were collected for various
35  analyses. We have re-written the sentence as "Samples included outcropping sediment layers

across the cliff face for grain size analyses, sediments with coating for geochemical analyses, and loess sediments for geochronological analysis".

5.      *Line 177: Suggestion to change to centimeter-scale*

This modification has been carried out.

6.      *Line 203: "Has previously been successful in coastal hydrogeophysical studies"*

This modification has been carried out.

7.      *Line 274: be explicit that this is groundwater, not surface, water flow*

This modification has been carried out.

8.      *Line 306: Indicate the components of the model that this would affect (permeability)*

The sentence has been rewritten as "Each scenario is modelled for two sandy gravel slopes with different permeabilities - one with a 0.5 m thick sand lens and the other with a 0.5 m thick gravel lens".

9.      *Line 308: So RocScience solves equation 2? Is there a relevant citation for this? The order of operations of the modeling here is a bit confusing. I assume Slide2 is solving equations 1 and 2.  Are equations 3-7 then being solved used a routine from Micallef et al 2020? What from this water modeling approach is being passed back to Slide2? If there are quantities being passed back to account for transient changes in pore fluid pressure and water flow, I cannot find those variables in the slope stability equation presented.  Please clarify.*

Slide2 solves equation 3 to solve equation 2, but unfortunately we have no relevant citation for this. Slide2 also solves equations 4 to 7 to model groundwater flow and estimate pore fluid pressure, which is separate from the slope stability equation. We have included the equations for pore water pressure, which were missing in the revised manuscript.

10.     *Line 319: Distribution of what quantity about the gullies? Their position?*

Here we are referring to their spatial distribution. This is now clarified in the text.

11. *Figure 2: Perhaps plot figure c in loglog space so data in the 1-10 and 10-100 range is more visible.*

This modification has been carried out.

12. *Line 332: In plan view*

This modification has been carried out.

13. *Line 341: Delete initial part of sentence*

We think it is important to include this information because the number of gullies increases later in the year.

14. *Line 371: Reorder sentence to refer to these in order.*

This modification has been carried out.

15. *Line 384: Specify uncertainty or say "approximately"*

We have included the word "approximately".

16. *Line 384: I find this statement confusing, I thought there were only two surveys, one in May and one in October?*

In section 3.2 we had specified the dates of the fourteen UAV surveys.

17. *Line 389: Replace with "occurred"*

This modification has been carried out.

18. *Figure 4: The positioning of these are confusing. I would suggest a left to right comparison instead as it is more intuitive to read that way. As I was going through I was initially confused why there was no erosion specified in some of the rows.*

This modification has been carried out.

19. *Figure 5: This figure has quite a lot of white space and the overlapping of precipitation and groundwater are hard to see. It might look better if you made this two panels so you could shrink the figure and not obscure the lab between the precipitation and groundwater data.*

I am not sure I understand what is being recommended. I have tried dividing the figure into two panels and it results in a larger figure with more white space. I thought it may be better to leave the figure as it is so that the readers can see that there was no gully erosion when rainfall intensity was less than 40 mm per day.

20. *Line 401: Can you quantify your volume uncertainty?*

The uncertainty is now included in the figure legend.

21. *Figure 6b: The quality of this graph is very low, and the peak storms are very hard to see. Perhaps a better strategy would be to report the mean and peak rainfall for each time interval between photos. This time series is not very informative as it is currently presented.*

The bars in the graph have been made more visible and the dates for the satellite images are now labelled.

22. *Figure 7: This figure is very confusing. Please try and clarify. Please give each individual element of this figure a different letter and reference. There are two A's and two B's which makes this very confusing. Please create legend items for each plot that include what the different colors of scatter points, vs. the smooth gold line are. Is the smooth gold a model or a best fit? May be useful to label the resistivity for sea water as you cite this as a marker for the model performance to be poor.*

These modifications have been carried out. We have also moved Figure 7b to Figure 8.

23. *Line 447: But how much spatial variability is meaningful? What is the minimum scale of variability that can be interpreted as a discrete lens of gravel or sand?*

The following text has been included: In general, due to lengthy signal-averaging times, ambient electromagnetic noise from the environment adds very small contribution to TDEM responses such that any along-profile variations are likely caused by geological heterogeneity. However, there is not a straightforward relationship between the magnitude of the TDEM voltage at any given time-gate and the resistivity within a particular subsurface volume. The situation becomes more complicated since the true Earth is characterised by multiscale heterogeneity such that spatial variations in the geology at all length scales superimpose their individual responses on one another to produce the final overall TDEM response that is measured. Thus, any analysis of the spatial variability of a time-gate plot, while informative, is largely qualitative and indicates only a first-order spatial distribution of causative subsurface structures.

24. *Line 449: It seems that this would be more appropriate with the actual time-gate data that you are showing in Figure 8. By keeping it in Figure 7, the reader will try and compare directly to other portions of figure 7, which may reduce the clarity of the manuscript.*

We have moved Figure 7b to Figure 8.

25. *Line 449: It seems that this would be more appropriate with the actual time-gate data that you are showing in Figure 8. By keeping it in Figure 7, the reader will try and compare directly to other portions of figure 7, which may reduce the clarity of the manuscript.*

We have moved Figure 7b to Figure 8.

26. *Figure 8: See previous comment of pairing Figure 7b with this figure. I do not think that insets are the clearest way to show this as it decreases the visualization of your time-gate profiles. I would suggest instead that you create a tiled plot with the image on the left and the associated time gate profiles on the right at a scale where they are easy to see. Please label the axes of the time gate figures. Please label Y axis of time gate graphs.*

These modifications have been carried out.

27. *Figure 9: Labels of these on the plots would make it easier for readers to differentiate at a glance. It would be really useful to see the time series of precipitation vs. the time series of Ff in addition to these plots, so readers can get a better idea of what the model is doing.*

*What time in the model run does this represent? Please add this as a legend to the plot as well (or put in a text label).*

These modifications have been carried out. The results shown are for the end of the simulation for each scenario. Unfortunately we were unable to generate the time series of precipitation vs. factor of safety in the time available.

28. *Please combine with previous figure, similar to next figure. Is plot C continuous data or discrete points?  Is this for the entire model run or the ending point?*

These modifications have been carried out. The data are discrete. The results shown are for the end of the simulation for each scenario.

29. *Figure 11: Include a plot similar to 10c? or remove 10c? I would recommend making these consistent. What are the red dots in this and the previous plot?  Please label and refer to in caption or remove.*

These modifications have been carried out. The red dots have been removed.

30. *Line 545: Where does this come from? I assume model output, but this is not mentioned or presented as a result of the study, so it is a bit confusing to talk about seepage flux here.  If the argument is that seepage flux is a key term in erosion (which I feel like it is), it may be useful to show us that model output prior to the discussion.*

This observation has now been moved to the Results section.

31. *Line 591: Where does this rainfall intensity threshold come from?  The modeling or the observational data. If this is from observational data, given the temporal spacing of your repeat surveys, how do you identify this threshold?*

We now explain that the threshold value is based on UAV and satellite imagery observations, which show that gullies form after rainfall events with an intensity higher than 40 mm per day (Figs. 5, 6), and the plot of factor of safety with rainfall intensity from the slope stability model for the first scenario ((I-D)3 for the slope with sand lens (Fig. 9).

**Comments made by Reviewer 1:**

215

32.  *Line 25ff: Note that the "OSL" ages in the paper are not actually optically stimulated luminescence derived ages. They are from related methods within the broader field of trapped-charge dating. I recommend replacing all references to OSL or Optically Stimulated Luminescence dating with the more general term: "luminescence dating"*
220 *(alternatively, you could also use infrared stimulated luminescence dating; IRSL)*

We have now replaced "optically stimulated luminescence dating" with "luminescence dating" everywhere. When the ages obtained in this study are discussed, we use "infrared stimulated luminescence ages".

225

33.  *Line 49: add a comma between "changes" and "such as"*

This modification has been carried out.

230

We hope that we have managed to address all your concerns and that you are satisfied with the above changes.

235  With best regards also on behalf of my co-authors,

Aaron Micallef

[revised manuscript text omitted]

Field Code Changed

| Page 18: [2] Formatted | ABC | 10/25/2020 9:54:00 PM |
|---|---|---|

Font: (Default) +Headings CS (Times New Roman), Complex Script Font: +Headings CS (Times New Roman)

| Page 18: [3] Formatted | ABC | 10/25/2020 9:54:00 PM |
|---|---|---|

Font: 10 pt, Complex Script Font: 10 pt

| Page 18: [4] Formatted | ABC | 10/25/2020 9:56:00 PM |
|---|---|---|

Space After:  0 pt, Line spacing:  1.5 lines

| Page 18: [5] Formatted | ABC | 10/25/2020 9:54:00 PM |
|---|---|---|

Font: (Default) +Headings CS (Times New Roman), 10 pt, Complex Script Font: +Headings CS (Times New Roman), 10 pt

| Page 18: [6] Formatted | ABC | 10/25/2020 9:54:00 PM |
|---|---|---|

Font: 10 pt, Complex Script Font: 10 pt

| Page 18: [7] Formatted | ABC | 10/25/2020 9:54:00 PM |
|---|---|---|

Font: (Default) +Headings CS (Times New Roman), 10 pt, Complex Script Font: +Headings CS (Times New Roman), 10 pt

| Page 18: [8] Formatted | ABC | 10/25/2020 9:54:00 PM |
|---|---|---|

Font: 10 pt, Complex Script Font: 10 pt

| Page 18: [9] Formatted | ABC | 10/25/2020 9:54:00 PM |
|---|---|---|

Font: (Default) +Headings CS (Times New Roman), 10 pt, Complex Script Font: +Headings CS (Times New Roman), 10 pt

| Page 18: [10] Formatted | ABC | 10/25/2020 9:56:00 PM |
|---|---|---|

Font: (Default) +Headings CS (Times New Roman), Complex Script Font: +Headings CS (Times New Roman)

| Page 18: [11] Formatted | ABC | 10/25/2020 9:56:00 PM |
|---|---|---|

Right, Space After:  0 pt, Line spacing:  1.5 lines

| Page 18: [12] Formatted | ABC | 10/25/2020 9:56:00 PM |
|---|---|---|

Font: (Default) +Headings CS (Times New Roman), Complex Script Font: +Headings CS (Times New Roman)

| Page 18: [13] Formatted | ABC | 10/25/2020 9:56:00 PM |
|---|---|---|

Font: (Default) +Headings CS (Times New Roman), Complex Script Font: +Headings CS (Times New Roman)

| Page 18: [14] Formatted | ABC | 10/25/2020 9:56:00 PM |
|---|---|---|

Font: 12 pt, Complex Script Font: 12 pt, Not Superscript/ Subscript

| Page 18: [15] Formatted | ABC | 10/25/2020 9:56:00 PM |
|---|---|---|

Font: 12 pt, Complex Script Font: 12 pt

| Page 18: [16] Formatted | ABC | 10/25/2020 9:56:00 PM |
|---|---|---|

Font: 10 pt, Complex Script Font: 10 pt

| Page 18: [17] Formatted | ABC | 10/25/2020 9:54:00 PM |
|---|---|---|

Font: 10 pt, Complex Script Font: 10 pt, Not Superscript/ Subscript

| Page 18: [18] Formatted | ABC | 10/25/2020 9:54:00 PM |
|---|---|---|

Font: (Default) +Headings CS (Times New Roman), 10 pt, Complex Script Font: +Headings CS (Times New Roman), 10 pt

| Page 18: [19] Formatted | ABC | 10/25/2020 9:56:00 PM |
|---|---|---|

Space Before:  0 pt, After:  0 pt, Line spacing:  1.5 lines

| Page 18: [20] Formatted | ABC | 10/25/2020 9:54:00 PM |
|---|---|---|

Font: (Default) +Headings CS (Times New Roman), 10 pt, Complex Script Font: +Headings CS (Times New Roman), 10 pt

| Page 18: [21] Formatted | ABC | 10/25/2020 9:59:00 PM |
|---|---|---|

Space Before:  0 pt, After:  0 pt, Line spacing:  1.5 lines

| Page 18: [22] Formatted | ABC | 10/25/2020 9:54:00 PM |
|---|---|---|

Font: 10 pt, Complex Script Font: 10 pt

| Page 18: [23] Formatted | ABC | 10/25/2020 9:54:00 PM |
|---|---|---|

Font: (Default) +Headings CS (Times New Roman), 10 pt, Complex Script Font: +Headings CS (Times New Roman), 10 pt

| Page 18: [24] Formatted | ABC | 10/25/2020 9:54:00 PM |
|---|---|---|

Font: (Default) +Headings CS (Times New Roman), 10 pt, Complex Script Font: +Headings CS (Times New Roman), 10 pt

| Page 18: [25] Formatted | ABC | 10/25/2020 9:54:00 PM |
|---|---|---|

Font: (Default) +Headings CS (Times New Roman), 10 pt, Complex Script Font: +Headings CS (Times New Roman), 10 pt

| Page 18: [26] Formatted | ABC | 10/25/2020 9:54:00 PM |
|---|---|---|

Font: 10 pt, Complex Script Font: 10 pt

| Page 18: [27] Formatted | ABC | 10/25/2020 9:54:00 PM |
|---|---|---|

Font: 10 pt, Complex Script Font: 10 pt

| Page 18: [28] Formatted | ABC | 10/25/2020 9:54:00 PM |
|---|---|---|

Font: (Default) +Headings CS (Times New Roman), 10 pt, Complex Script Font: +Headings CS (Times New Roman), 10 pt

| Page 18: [29] Formatted | ABC | 10/25/2020 9:54:00 PM |
|---|---|---|

Font: (Default) +Headings CS (Times New Roman), 10 pt, Complex Script Font: +Headings CS (Times New Roman), 10 pt

| Page 18: [30] Formatted | ABC | 10/25/2020 9:56:00 PM |
|---|---|---|

Right, Space Before: 0 pt, After: 0 pt, Line spacing: 1.5 lines

| Page 18: [31] Formatted | ABC | 10/25/2020 9:56:00 PM |
|---|---|---|

Font: (Default) +Headings CS (Times New Roman), Complex Script Font: +Headings CS (Times New Roman)

| Page 18: [32] Formatted | ABC | 10/25/2020 9:56:00 PM |
|---|---|---|

Font: (Default) +Headings CS (Times New Roman), Complex Script Font: +Headings CS (Times New Roman)

| Page 18: [33] Formatted | ABC | 10/25/2020 9:56:00 PM |
|---|---|---|

Font: (Default) +Headings CS (Times New Roman), Complex Script Font: +Headings CS (Times New Roman)

| Page 18: [34] Formatted | ABC | 10/25/2020 9:56:00 PM |
|---|---|---|

Font: 12 pt, Complex Script Font: 12 pt

| Page 18: [35] Formatted | ABC | 10/25/2020 9:56:00 PM |
|---|---|---|

Font: 10 pt, Complex Script Font: 10 pt

| Page 18: [36] Formatted | ABC | 10/25/2020 9:54:00 PM |
|---|---|---|

Font: (Default) +Headings CS (Times New Roman), 10 pt, Complex Script Font: +Headings CS (Times New Roman), 10 pt

| Page 18: [37] Formatted | ABC | 10/25/2020 9:56:00 PM |
|---|---|---|

Space Before: 0 pt, After: 0 pt, Line spacing: 1.5 lines

**Page 18: [38] Formatted**            **ABC**            **10/25/2020 9:54:00 PM**

Font: (Default) +Headings CS (Times New Roman), 10 pt, Complex Script Font: +Headings CS (Times New Roman), 10 pt

**Page 18: [39] Formatted**            **ABC**            **10/25/2020 9:56:00 PM**

Right, Space Before: 0 pt, After: 0 pt, Line spacing: 1.5 lines

**Page 18: [40] Formatted**            **ABC**            **10/25/2020 9:57:00 PM**

Font: (Default) +Headings CS (Times New Roman), Complex Script Font: +Headings CS (Times New Roman)

**Page 18: [41] Formatted**            **ABC**            **10/25/2020 9:57:00 PM**

Font: (Default) +Headings CS (Times New Roman), Complex Script Font: +Headings CS (Times New Roman)

**Page 18: [42] Formatted**            **ABC**            **10/25/2020 9:54:00 PM**

Font: 10 pt, Complex Script Font: 10 pt

**Page 18: [43] Formatted**            **ABC**            **10/25/2020 9:54:00 PM**

Font: (Default) +Headings CS (Times New Roman), 10 pt, Complex Script Font: +Headings CS (Times New Roman), 10 pt

**Page 18: [44] Formatted**            **ABC**            **10/25/2020 9:54:00 PM**

Font: 10 pt, Complex Script Font: 10 pt

**Page 18: [45] Formatted**            **ABC**            **10/25/2020 9:54:00 PM**

Font: (Default) +Headings CS (Times New Roman), 10 pt, Complex Script Font: +Headings CS (Times New Roman), 10 pt